# PHYSICS-ALIGNED FIELD RECONSTRUCTION WITH DIFFUSION BRIDGE

**Zeyu Li, Hongkun Dou, Shen Fang, Wang Han, Yue Deng, Lijun Yang**[*]
Beihang University
{lizeyu123478, douhk, eureka10shen,
drwanghan, ydeng, yanglijun}@buaa.edu.cn

## ABSTRACT

The reconstruction of physical fields from sparse measurements is pivotal in both scientific research and engineering applications. Traditional methods are increasingly supplemented by deep learning models due to their efficacy in extracting features from data. However, except for the low accuracy on complex physical systems, these models often fail to comply with essential physical constraints, such as governing equations and boundary conditions. To overcome this limitation, we introduce a novel data-driven field reconstruction framework, termed the Physics-aligned Schrödinger Bridge (PalSB). This framework leverages a diffusion bridge mechanism that is specifically tailored to align with physical constraints. The PalSB approach incorporates a dual-stage training process designed to address both local reconstruction mapping and global physical principles. Additionally, a boundary-aware sampling technique is implemented to ensure adherence to physical boundary conditions. We demonstrate the effectiveness of PalSB through its application to three complex nonlinear systems: cylinder flow from Particle Image Velocimetry experiments, two-dimensional turbulence, and a reaction-diffusion system. The results reveal that PalSB not only achieves higher accuracy but also exhibits enhanced compliance with physical constraints compared to existing methods. This highlights PalSB's capability to generate high-quality representations of intricate physical interactions, showcasing its potential for advancing field reconstruction techniques. The source code can be found at https://github.com/lzy12301/PalSB.

## 1 INTRODUCTION

Field reconstruction is critically important in several domains, including fluid mechanics (Fukami et al., 2023; Manohar et al., 2018), meteorology (Carrassi et al., 2018; Kondrashov & Ghil, 2006; Tello Alonso et al., 2010), and astrophysics (The Event Horizon Telescope Collaboration, 2019), where high-fidelity data is essential. This process aims to recover the spatiotemporal information of a physical system from limited observations gathered by sensors. The challenges of measurement sparsity and noise necessitate efficient and accurate methodologies that enhance understanding of complex systems beyond the resolution capabilities of the instruments used.

However, the intrinsic challenges posed by the ill-posed nature of inverse problems and the complex spatiotemporal interactions within many systems, such as turbulence, render these reconstructions particularly difficult (Buzzicotti, 2023). Traditional physics-based methods, which repeatedly recalibrate to balance observational data with physical laws, often incur significant computational costs due to their intensive requirements for high-fidelity simulations.

Recent advances in machine learning, especially diffusion-based models, have shown promise in managing complex data distributions and have been successfully applied in diverse areas such as image translation (Zhang et al., 2023; Luo et al., 2023; Meng et al., 2022; Su et al., 2023; Yue et al., 2023), molecular generation (Xu et al., 2022; Watson et al., 2023; Igashov et al., 2024), and dynamic forecasting (Gao et al., 2023; Yoon et al., 2023; Cachay et al., 2023). Despite achieving remarkable point-wise accuracy compared to other end-to-end methods, these models often fail to

---
[*]Corresponding author

align with physical laws in the context of physical field reconstruction. The challenge is exacerbated by the non-convex and often intractable nature of the domains defined by the governing physical laws, such as partial differential equations (PDEs). The projection-based methods for enforcing hard constraints (Liu et al., 2023a; Lou & Ermon, 2023; Christopher et al., 2024) are therefore inapplicable due to the intractable domain of constraints. An alternative approach involves embedding these physical laws directly into the optimization objective as soft constraints, as seen in Physics-Informed Neural Networks (PINNs) (Raissi et al., 2019; 2020). However, this strategy can lead to severe convergence issues without appropriate initial conditions, due to the non-convex nature of these objectives (Krishnapriyan et al., 2021; Wang et al., 2022a).

In response, we propose a **P**hysics-**al**igned **S**chrödinger **B**ridge (PalSB) framework to address these challenges, ensuring both efficient and physically compliant field reconstruction. Our framework integrates a two-stage training strategy that first uses a diffusion Schrödinger bridge (DSB) for high-quality field generation from sparse measurements. This initial stage creates super-resolved fields which, while accurate, may not fully comply with physical laws. With this as a foundation, the model is further refined in the second stage through a physics-informed objective tailored to such a diffusion bridge, enhancing its adherence to physical principles. Additionally, we innovate the sampling process to ensure boundary condition compliance and to streamline generation, targeting efficiency within 10 number of function evaluations (NFEs).

Our main contributions can be summarized as follows:

- We explore the application of DSB in physical field reconstruction from sparse measurements, focusing on efficient and scalable training strategies that circumvent the need for full field data during initial training phases.
- We develop a physics-aligned fine-tuning approach for generative models to address optimization challenges associated with physics-informed loss functions, significantly improving the physical compliance of the generated fields.
- We introduce an innovative sampling technique that effectively incorporates boundary conditions into the generative process.

## 2 BACKGROUND

### PROBLEM SETUPS

Given the low-fidelity measurements denoted as $\mathbf{y} \in \mathbb{R}^m$ discretized over domain $\Omega \subset \mathbb{R}^d$, the goal of field reconstruction is to recover the high-fidelity spatiotemporal field $\mathbf{x}_0 \in \mathbb{R}^n$ over the same domain. The forward mapping from $\mathbf{x}_0$ to $\mathbf{y}$ can be represented as $\mathbf{y} = \mathcal{H}(\mathbf{x}_0) + \epsilon$, where $\mathcal{H}$ is the observation operator and $\epsilon$ is the Gaussian noise introduced by measurement errors. In other words, the measurements can be regarded as a sample of conditional distribution $p(\mathbf{y}|\mathbf{x}_0) = \mathcal{N}(\mathbf{y}; \mathcal{H}(\mathbf{x}_0), \sigma_\epsilon^2 \mathbf{I}_m)$. To inversely acquire $\mathbf{x}_0$ from measurements $\mathbf{y}$, one need to model the posterior probability $p(\mathbf{x}_0|\mathbf{y})$ whose form is inaccessible in most scenarios. From either a paired dataset $\left\{ \mathbf{x}_0^{(i)}, \mathbf{y}^{(i)} \right\}_{i=1}^N$ or a given prior distribution of $\mathbf{x}_0$, such posterior can be statistically approximated in a data-driven manner.

### DIFFUSION SCHRÖDINGER BRIDGE

The Schrödinger bridge problem, stemming from optimal transport and stochastic processes (Schrödinger, 1932; Léonard, 2012; Chen et al., 2016), provides a methodological framework to efficiently transition between two probability distributions $p_0$ and $p_1$. This is achieved through the formulation of forward and backward stochastic differential equations (SDEs):

$$d\mathbf{x}_t = [\mathbf{f}(t) + \beta(t)\nabla \log \Psi(\mathbf{x}_t, t)]dt + \sqrt{\beta(t)}dW_t \tag{1a}$$

$$d\mathbf{x}_t = [\mathbf{f}(t) - \beta(t)\nabla \log \hat{\Psi}(\mathbf{x}_t, t)]dt + \sqrt{\beta(t)}d\overline{W}_t \tag{1b}$$

where $t \in [0, 1]$ is the time variable, $\mathbf{f}(t)$ represents the drift term at time $t$, $\beta(t)$ is the diffusion coefficient, and $dW_t$ and $d\overline{W}_t$ are the incremental Wiener processes for the forward and backward SDEs,

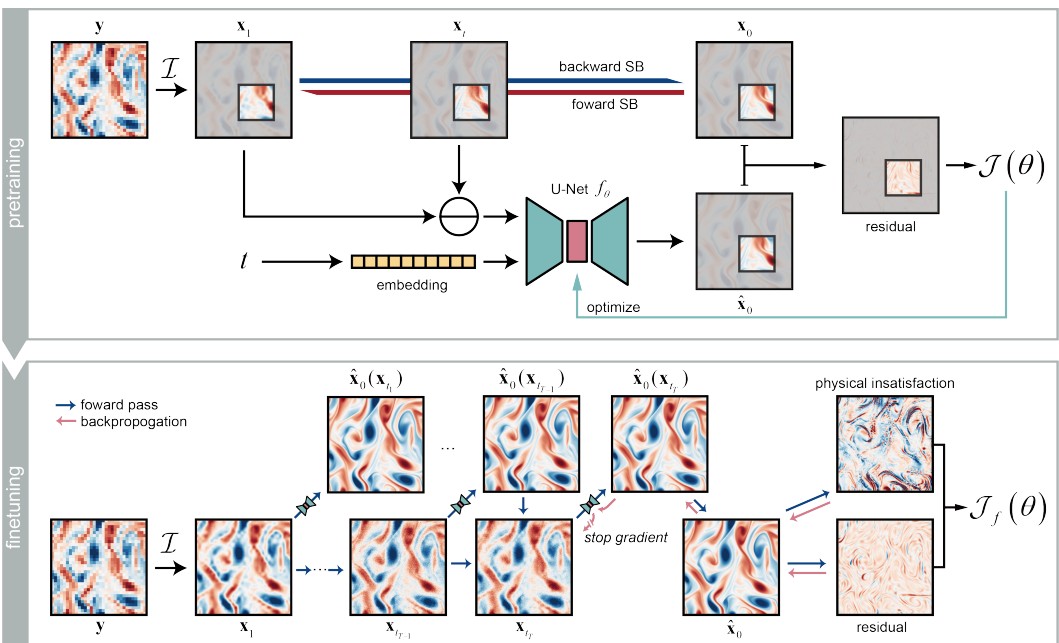

Figure 1: Work flow of PalSB. In pretraining stage (top row), the low-fidelity field $\mathbf{y}$ is first interpolated to the same grid for output and the Gaussian-perturbed linear interpolation of the paired samples is then fed into the neural network to make a prediction of the high-fidelity output, where the residual between predition and label is utilized to optimize the neural network. In finetuning stage (bottom row), leveraging the pretrained model, a prediction of high-fidelity field is sampled from the low-fidelity condition through the DSB, assessed then by two metrics that evaluate the physical loss and regression loss. Subsequently, the model is tuned through the sampling path using the weighted loss.

respectively. The functions $\Psi(\mathbf{x}_t, t)$ and $\hat{\Psi}(\mathbf{x}_t, t)$ are the forward and backward potentials guiding the transformation between $p_0$ and $p_1$. Modifying the drift term $\mathbf{f}(t)$ to $\mathbf{f}(t)' - \beta(t)\nabla \log \Psi(\mathbf{x}_t, t)$ integrates the formulation into the framework of score-based generative models (Song et al., 2021), where the score function $\nabla \log p_t(\mathbf{x}, t) = \nabla \log[\Psi(\mathbf{x}, t)\hat{\Psi}(\mathbf{x}, t)]$ guides the evolution of the probability from data distribution $p_0$ to a zero-mean Gaussian $p_1$. Considering that the diffusion process in a DSB does not necessarily conclude at a Gaussian distribution, additional information can be injected into the final distribution $p_1$. When the starting and ending distributions are $p_0(\mathbf{x})$ and $p(\mathbf{y}|\mathbf{x}_0)$, respectively, the forward SDE models the degradation from high-fidelity to low-fidelity data, which can theoretically be reversed via the backward SDE once the function $\hat{\Psi}(\mathbf{x}, t)$ is determined.

Compared to vanilla diffusion models such as DDPM (Ho et al., 2020) and score SDE (Song et al., 2021), DSB is more flexible in alternating the boundary distributions, promoting the direct modeling of inverse problem without starting from pure Gaussian noise. Moreover, the optimality of DSB in transferring distributions can significantly reduce the number of sampling steps. Despite these advantages, solving the Schrödinger bridge problem is complex in practice since a close-form solution for DSB does not exist in general cases. Some methods approximate the solution using an iterative algorithm that needs to simulate the learned SDE (De Bortoli et al., 2021; Chen et al., 2022; Tang et al., 2024), while some recent studies introduce the simulation-free approaches for solving DSB (Tong et al., 2024; Liu et al., 2023b). Among these methods, the I$^2$SB approach (Liu et al., 2023b), which is simple for implementation, proposes that both potentials conform to the constraints of the probabilistic density function, resulting in corresponding score functions $\nabla \log \Psi(\mathbf{x}_t, t)$ and $\nabla \log \hat{\Psi}(\mathbf{x}_t, t)$ for the reversed paths of the described SDEs:

$$d\mathbf{x}_t = \mathbf{f}(t)dt + \sqrt{\beta(t)}dW_t, \mathbf{x}_0 \sim \hat{\Psi}(\cdot, 0) \tag{2a}$$

$$d\mathbf{x}_t = \mathbf{f}(t)dt + \sqrt{\beta(t)}d\overline{W}_t, \mathbf{x}_1 \sim \Psi(\cdot, 1) \tag{2b}$$

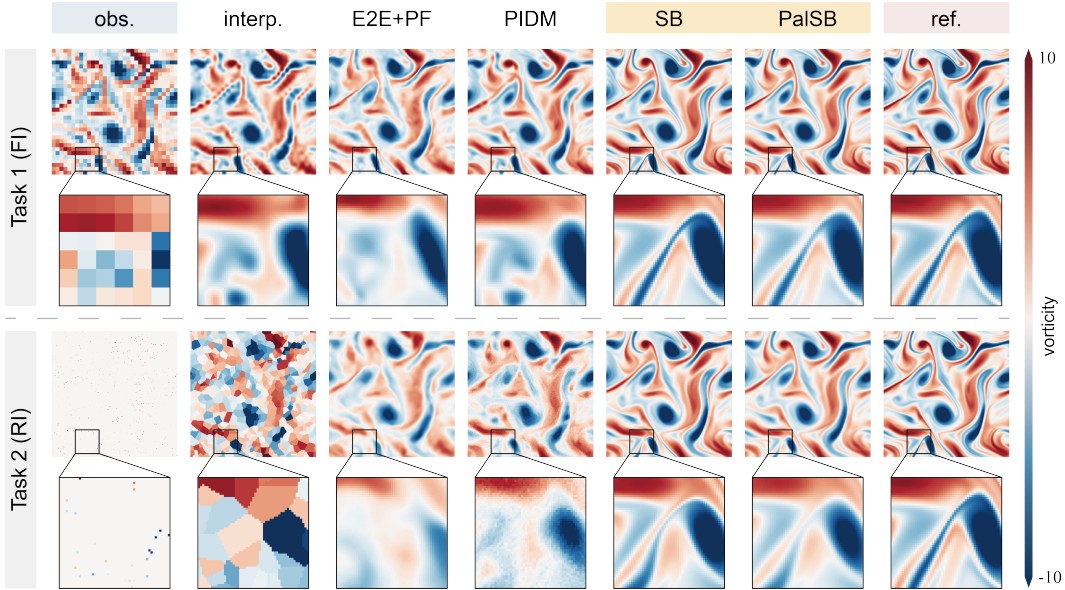

Figure 2: Visual comparison between different methods on the two varying tasks. In the first task (FI, top row), the low-fidelity observation (denoted as obs.) is 8x down-sampled from high-fidelity field of Kolmogorov flow on 256×256 grid. In the second task (RI, bottom row), the observation is randomly sampled from high-fidelity field with 99% of the field masked. Our proposed method (SB and PalSB) visually outperforms other baselines that better recovers the spatial patterns as compared to the reference (denoted as ref.)

Given appropriate boundary distributions, the posterior distributions $\Psi(\mathbf{x}_t, t|\mathbf{x}_0)$ and $\hat{\Psi}(\mathbf{x}_t, t|\mathbf{x}_1)$ become accessible. The posterior distribution is then defined by Nelson's duality (Liu et al., 2023b):

$$p_t(\mathbf{x}_t, t|\mathbf{x}_0, \mathbf{x}_1) = \mathcal{N}\left(\mathbf{x}_t; \frac{\bar{\sigma}_t^2}{\bar{\sigma}_t^2 + \sigma_t^2}\mathbf{x}_0 + (1 - \frac{\bar{\sigma}_t^2}{\bar{\sigma}_t^2 + \sigma_t^2})\mathbf{x}_1, \frac{\bar{\sigma}_t^2 \sigma_t^2}{\bar{\sigma}_t^2 + \sigma_t^2}\mathbf{I}\right) \tag{3}$$

where $\sigma_t^2 = \int_0^t \beta(\tau)d\tau$ and $\bar{\sigma}_t^2 = \int_t^1 \beta(\tau)d\tau$ represent the accumulated variances. The DSB model is then trained using paired data $(\mathbf{x}_0, \mathbf{x}_1)$ in a Denoising Diffusion Probabilistic Model (DDPM) style (Ho et al., 2020), with a training objective defined as:

$$\mathcal{J}(\theta) = \|\epsilon_\theta(\mathbf{x}_t, t) - \frac{\mathbf{x}_t - \mathbf{x}_0}{\sigma_t}\| \tag{4}$$

where $\mathbf{x}_t$ is sampled from the analytic posterior distribution in equation 3.

## 3 METHODOLOGY

The pipeline of PalSB, as represented in Fig. 1, includes a two-stage (pretraining and finetuning) training process and a particularly designed sampling process.

### 3.1 DATA-DRIVEN PRETRAINING

In the realm of field reconstruction, the boundary distributions for the DSB in Eq. 1 are characterized by the distribution of high-fidelity data, $p_0$, and the distribution of corrupted data, $p_1$. The corrupted data samples are derived from high-fidelity samples through the equation:

$$\mathbf{x}_1 = \mathcal{I}(\mathbf{y}) = \mathcal{I}(\mathcal{H}(\mathbf{x}_0) + \epsilon), \quad \mathbf{x}_0 \sim p_0, \quad \epsilon \sim \mathcal{N}(\mathbf{0}, \sigma_\epsilon^2 \mathbf{I}_m) \tag{5}$$

where $\mathcal{I}$ denotes a predefined interpolation function (e.g., nearest, bi-linear, or bi-cubic) that up-scales the sparse observations to match the dimensions of high-fidelity samples.

We find that training directly on high-resolution fields dramatically slows the convergence rate and increases the training time for each iteration. To tackle such problem, inspired by the patch-based training widely used in image restoration (Yang et al., 2019), we train the model on small, randomly cropped local patches from high-resolution field, combined with a scalable network for further inference on global field. This strategy not only focuses on capturing the local structure of the physical field but also mitigates the computational burden associated with processing high-resolution inputs. We then reformulate the training objective to directly parameterize the super-resolving operator, as per the following equation:

$$\mathcal{J}(\theta) = \mathbb{E}_t \mathbb{E}_{\tilde{\mathbf{x}}_0, \tilde{\mathbf{x}}_1 \sim \text{crop}(\mathbf{x}_0, \mathbf{x}_1)} \mathbb{E}_{\mathbf{x}_0, \mathbf{x}_1 \sim p_0, p_1} \left[ \| f_\theta(\tilde{\mathbf{x}}_t, \tilde{\mathbf{x}}_1, t) - \tilde{\mathbf{x}}_0 \|^2 \right] \tag{6}$$

Here, $f_\theta$ represents the field prediction network, defined as $f_\theta = -\sigma_t \epsilon_\theta + \mathbf{x}_t$. This network is designed to accommodate inputs of varying resolutions, facilitating scalability to higher-resolution inputs (Luo et al., 2023). The pretraining procedure, including this network's deployment, is detailed in Alg. 1, with further implementation specifics discussed in Appendix A.1.

## 3.2 Physics-aligned Finetuning (PF)

In contrast to conventional image or video generation, the simulation of physical fields needs to take physical laws into consideration, typically expressed as PDEs. These laws are represented by the constraint $\mathcal{F}(\mathbf{x}) = 0$, where $\mathcal{F}$ quantifies deviations from physical laws in a sample.

Due to the spatiotemporal discretization of the generated sample, perfect conformity with these constraints is challenging to achieve. Otherwise, the most straightforward way is to apply physics-informed losses to the one-step prediction $\| \mathcal{F}(f_\theta(\tilde{\mathbf{x}}_t, \tilde{\mathbf{x}}_1, t)) \|$ as an additional penalty in Eq. 6. However, differing from learning an end-to-end model, the score matching objective in Eq. 6 leads to an expectation (i.e., not a specific sample) over the Gaussian noise on the DSB, which is a single-step rough prediction of the target field. Accordingly, directly optimizing physics-informed loss on such inaccurate predictions hinders the accurate convergence of the training dynamics.

Instead, drawing inspiration from reinforcement learning with human feedback (RLHF) for diffusion models (Lee et al., 2023; Fan et al., 2024), our approach seeks to minimize these deviations on the generated samples throughout the generating path by finetuning the model parameters, $\theta$. The proposed physics-aligned finetuning (PF) objective is formulated as follows:

$$\mathcal{J}_f(\theta) = \mathbb{E}_\mathbf{y} \mathbb{E}_{\hat{\mathbf{x}}_0 \sim p_\theta(\mathbf{x}_0 | \mathbf{y})} \left[ \gamma_{phys} \| \mathcal{F}(\hat{\mathbf{x}}_0) \| + \gamma_{reg} \| \hat{\mathbf{x}}_0 - \mathbf{x}_0 \| \right] \tag{7}$$

In this equation, $\hat{\mathbf{x}}_0$ represents a sample generated along the sampling path, while $\gamma_{phys}$ and $\gamma_{reg}$ are hyperparameters that balance the loss components. Since the ground truth is available, we introduce an additional regression loss (the second term on the right-hand side of equation 7), which acts as a regularization term. This helps maintain the stability of the optimization process and prevents the model from collapsing into a trivial solution, especially when the underlying constraints are ill-posed. This additional term is conceptually similar to the KL-regularization used in DPOK (Fan et al., 2024), which constrains the tuned distribution to remain close to the pretrained distribution. However, we directly utilize the pretraining samples $\mathbf{x}_0$, whereas DPOK relies on model-generated samples.

Since the whole objective in Eq. 7 is differentiable w.r.t. the model's parameters, we apply a gradient-based method to directly optimize the finetuning objective. The computational graph of the generating process is unrolled to compute the gradients:

$$\nabla_\theta \mathcal{J}_f(\theta) = \frac{\partial \mathcal{J}_f}{\partial \theta} + \sum_{i=1}^{T} \left( \frac{\partial \mathbf{x}_{t_i}}{\partial \theta} \right)^\top \frac{\partial \mathcal{J}_f}{\partial \mathbf{x}_{t_i}} \tag{8}$$

The indices $0 < t_T < t_{T-1} < ... < t_1 = 1$ represent the diffusion steps used for sample generation, with $\top$ indicating the transpose of the Jacobian matrix. Due to the potentially high memory demand of unrolling the full computational graph, backpropagation is truncated for all steps $i < T$, as suggested

in the literature (Prabhudesai et al., 2023; Clark et al., 2023). The specifics of this finetuning process are detailed in Alg. 2 and further discussed in Appendix A.1.

## 3.3 SAMPLING STRATEGY

Training focused on local features ensures model scalability and data efficiency but may inadvertently neglect the global coherence necessary for physical field reconstructions, particularly with respect to boundary conditions. Moreover, the inherent scalability of neural networks typically restricts them to learning local mappings, necessitating a strategy to integrate global features via the learned local mappings. To address these challenges and enhance the speed of the sampling process, we introduce two simple yet effective techniques:

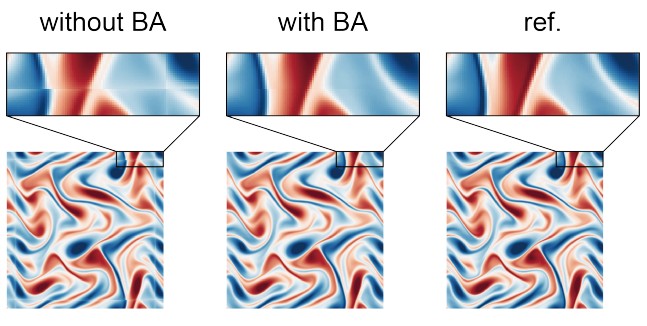

without BA          with BA          ref.

Figure 3: Efficacy of boundary-aware sampling strategy

**Boundary-aware sampling (BA).** Initially, as illustrated in Figure 4, boundary-aware (BA) sampling begins by padding the low-fidelity input, $\mathbf{y}$, according to the specific type of boundary conditions. Subsequently, the padded input is interpolated to generate the practical input $\mathbf{x}_1$ used during inference: $\mathbf{x}_1 = \mathcal{I}\left(\text{pad}(\mathbf{y})\right)$. Notably, this approach is not employed during training because the patch-based training methodology inherently disrupts the original boundary conditions.

**Early-stop sampling (ES).** As a secondary enhancement, we find that utilizing smaller step sizes combined with an early stop strategy significantly improves performance compared to traditional sampling methods that use larger step sizes for the same NFEs (Fig. 5). Remarkably, in some instances, even single-step generation achieves performance comparable to the multi-step generation, as demonstrated in Appendix A.1 and Fig. 7.

## 4 EXPERIMENTAL SETUP

### 4.1 DATASETS

**Cylinder flow measured by PIV.** Under specific circumstance, the fluid flowing around a blund body (e.g., a cylinder) can induce periodically falling vortices, which is governed by the Navier-Stokes equations quantifying the conservation laws of mass, momentum and energy. The data are gathered from the particle image velocimetry (PIV) experiment, including only a single trajectory of velocity vectors with total length of 879 frames and original resolution of $50 \times 67$, in which the former 70% of the trajectory is used for training and the rest is used for test.

**2D forced turbulence.** Kolmogorov flow (Kolmogorov et al., 1997; Boffetta & Ecke, 2012; Chandler & Kerswell, 2013), a canonical system in studying 2D homogeneous isotropic turbulence (HIT) in fluid dynamics, represents more complicated spatiotemporal patterns controlled by the incompressible Navier-Stokes equations. We use the high-fidelity dataset with spatial resolution of $256 \times 256$ published in (Shu et al., 2023), which contains 40 trajectories with 320 frames in each trajectory. We use 90% of the trajectories as the training data and the rest 10% as the test data.

**Reaction-diffusion system.** Reaction-diffusion system of the Gray-Scott type (simply denoted as RDGS) is another nonliear dynamical system widely tackled in many biological (Maini et al., 2004) and chemical applications (Vervloet et al., 2012). The data used here is simulated on a $256 \times 256$ spatial grid with periodic boundary condition. Two trajectories with 3000 frames that start with different initial conditions are used for training and testing, respectively. Notably, with only one trajectory available for training, the amount and diversity of this dataset is much less than the dataset for Kolmogorov flow.

Detailed descriptions of the datasets and the corresponding physical constraints can be found in A.3. Beyond these 2D cases, PalSB can also work on 3D system. The corresponding description and results can be found in B.1

## 4.2 TASKS

For each dataset, we train and test the model on two practical tasks in field reconstruction: **(1) Fourier-space interpolation (FI (Buzzicotti, 2023)) and (2) real-space interpolation (RI (Buzzicotti, 2023)).** The former is commonly seen in image-based measurements such as PIV and BOS (Background Oriented Schlieren method), where the physical field cannot be simultaneously acquired at high resolution and covering a large field of view, resulting in a great demand for accurate and data-efficient super-resolution method. On the other hand, the latter usually involves in the intrusive measurements such as hot-wire velocimetry and Pitot, which can interact with the physical field itself, leading to a restricted number of sensor arrangements.

## 4.3 BASELINES

We choose the advanced methods in physical field reconstruction as baselines, including the conventional interpolation method, the End-to-End model based-on FNO (Li et al., 2020) and PIDM (Shu et al., 2023). Detailed description of the baselines can be found in Appendix A.2.

**Interpolation.** Conventional approach for filling in missing data for field reconstruction from sparse data. In our experiments, bi-cubic interpolation is used for the FI task and nearest interpolation is used for the RI task.

**End-to-End model with physics-aligned finetuning.** Typical style of supervised learning for data-driven reconstruction of field, directly mapping the low-fidelity data to match with the high-fidelity labels. Here, we choose FNO (Li et al., 2020) as the backbone of the neural network, which is suitable for this task (Takamoto et al., 2022). To make fair comparison, we also apply the physics-informed loss function to finetune the end-to-end model.

**PIDM (Shu et al., 2023).** With a denoising network pretrained on high-fidelity data as the prior distribution, DDPM can generate the reconstructed field using proper techniques for injecting posterior information such as SDEdit (Meng et al., 2022). In PIDM, by conditioning on the gradient of the equation residual of the input field, the physical insatisfaction of generated contents can be implicitly reduced in some cases.

**SB**. SB method is the non-finetuned version of PalSB, which is equipped with the same techniques used by PalSB except for the physics-aligned finetuning.

## 4.4 EVALUATION METRICS

We use two types of metric that can evaluate the point-wise accuracy and physical insatisfaction, respectively.

**nRMSE** refers to the normalized relative mean squared error (Raissi et al., 2020), which evaluate the relative L2 error between the reference field and the predicted field. **MSE** refers to the mean squared error (L2 error). **MAE** refers to the mean absolute error (L1 error). **Correlation** refers to the Pearson correlation coefficient. These metrics evaluates the point-wise accuracy of the predicted fields

**nER** refers to the normalized equation residual (Shu et al., 2023), evaluating the corresponding physical insatisfaction of the given field.

## 5 RESULTS

### 5.1 COMPARISON WITH BASELINES

Figures 2 provides a qualitative comparison between the proposed Physics-aligned Schrödinger Bridge (PalSB) framework and existing baselines, highlighting our framework's superior ability to accurately recover spatial patterns in Kolmogorov flow across two distinct tasks: FI and RI. Additional comparison between the methods is shown in Figs. 12-15.

Quantitatively, Table 1 displays the metrics used to evaluate the point-wise accuracy and equation satisfaction of the predicted fields from various models. PalSB shows significant improvements over other state-of-the-art methods across all three physical systems and both tasks. Notably, even when finetuned using the physics-informed loss as in Eq. 7, the E2E model exhibits higher deviations from the physical laws, particularly in systems governed by highly nonlinear equations such as the Kolmogorov flow and the reaction-diffusion system. This can be attributed to challenges in poor initialization of the weights and the tendency of the optimization through a non-convex physics-informed loss to settle into suboptimal local minima, leading to potential convergence failures (Krishnapriyan et al., 2021; Wang et al., 2022a).

Furthermore, the PIDM, while implicitly conditioned on equation residual information, only partially reduces physical dissatisfaction in the Kolmogorov flow and fails in other tests. This underscores the limitations of the DDPM framework in scenarios with insufficient data. The spatiotemporally scattered measurements in the RI task can significantly disrupt the dynamics, as evidenced by the nER metric comparison between FI and RI for the interpolation method in Table 2. This disruption hinders reconstruction through SDEdit that starts from the interpolated field, resulting in poorer performance of PIDM in the RI task.

Moreover, introducing an additional 5% Gaussian noise to the input low-fidelity data only marginally impacts PalSB's performance, demonstrating its robustness in handling noisy observations (detailed descriptions are available in Appendix B.2).

Table 1: Performance comparison of models across different cases and tasks. The metrics with blue color evaluate the errors with the reference data, while the metric with red color evaluates the physics compliance.

| Case | Task | Model | nRMSE ↓ | MSE ↓ | MAE ↓ | Correlation ↑ | nER ↓ |
|------|------|-------|---------|-------|-------|---------------|-------|
| Cy. Flow | FI | Interp. | 0.274 | 2.29E-05 | 3.06E-03 | 0.843 | 5.90E-02 |
| | | E2E+PF | 0.094 | 3.42E-06 | 1.08E-03 | 0.972 | 8.32E-03 |
| | | PIDM | 0.261 | 2.14E-05 | 3.05E-03 | 0.766 | 1.60E-02 |
| | | SB | 0.063 | **1.36E-06** | 7.18E-04 | **0.994** | 1.50E-02 |
| | | PalSB | **0.062** | 1.42E-06 | **7.04E-04** | 0.974 | **1.22E-03** |
| | RI | Interp. | 0.301 | 2.80E-05 | 3.10E-03 | 0.898 | 1.70E+00 |
| | | E2E+PF | 0.100 | 3.81E-06 | 1.17E-03 | 0.906 | 9.48E-03 |
| | | PIDM | 0.121 | 4.97E-06 | 1.45E-03 | 0.894 | 4.50E-02 |
| | | SB | 0.092 | 2.92E-06 | 1.04E-03 | **0.972** | 2.10E-02 |
| | | PalSB | **0.090** | **2.75E-06** | **1.02E-03** | 0.955 | **8.74E-04** |
| Kol. Flow | FI | Interp. | 0.538 | 6.58E-01 | 1.88E+00 | 0.857 | 1.40E-01 |
| | | E2E+PF | 0.288 | 1.91E-01 | 1.04E+00 | 0.959 | 3.91E-01 |
| | | PIDM | 0.512 | 5.95E-01 | 1.77E+00 | 0.871 | 3.45E-01 |
| | | SB | **0.077** | **1.41E-01** | **2.24E-01** | 0.996 | 7.99E-02 |
| | | PalSB | 0.081 | 1.56E-01 | 2.66E-01 | **0.997** | **2.40E-02** |
| | RI | Interp. | 0.582 | 7.69E-01 | 1.88E+00 | 0.841 | 6.22E+02 |
| | | E2E+PF | 0.466 | 5.00E-01 | 1.72E+00 | 0.908 | 2.68E+00 |
| | | PIDM | 0.381 | 3.31E-01 | 1.34E+00 | 0.934 | 2.73E+01 |
| | | SB | **0.230** | **1.23E-01** | **7.16E-01** | **0.973** | 1.95E+00 |
| | | PalSB | 0.253 | 1.48E+00 | 8.83E-01 | **0.973** | **5.65E-01** |
| RDGS | FI | Interp. | 0.282 | 2.59E-02 | 1.01E-01 | 0.864 | 3.68E-05 |
| | | E2E+PF | 0.261 | 2.18E-02 | 1.03E-01 | 0.883 | 8.17E-05 |
| | | PIDM | 0.268 | 2.30E-02 | 1.06E-01 | 0.870 | 1.23E-03 |
| | | SB | 0.107 | 4.25E-03 | 2.66E-02 | 0.980 | 1.58E-06 |
| | | PalSB | **0.100** | **3.76E-03** | **2.46E-02** | **0.981** | **8.91E-08** |
| | RI | Interp. | 0.292 | 2.78E-02 | 9.72E-02 | 0.855 | 2.95E-02 |
| | | E2E+PF | 0.394 | 3.21E-02 | 1.25E-01 | 0.883 | 5.37E-05 |
| | | PIDM | 0.277 | 1.54E-02 | 9.18E-02 | 0.918 | 3.83E-03 |
| | | SB | 0.194 | **1.34E-02** | **5.76E-02** | **0.942** | 1.86E-06 |
| | | PalSB | **0.193** | 1.35E-02 | 5.79E-02 | 0.941 | **1.69E-07** |

## 5.2 ABLATION STUDIES

Our ablation studies, summarized in Table 2, investigate the effectiveness of specific designed modules within the PalSB framework, with the number of sampling steps fixed at 10. Notably, our

Table 2: Ablation studies across different cases and tasks. The metrics with blue color evaluate the errors with the reference data, while the metric with red color evaluates the physics compliance.

| Case | Task | Model | nRMSE ↓ | MSE ↓ | MAE ↓ | Correlation ↑ | nER ↓ |
|------|------|-------|---------|-------|-------|---------------|-------|
| Cy. Flow | FI | Full model | **0.062** | 1.42E-06 | **7.04E-04** | 0.974 | **1.22E-03** |
| | | w/o PF | 0.063 | **1.36E-06** | 7.18E-04 | **0.994** | 1.54E-02 |
| | | w/o DS | **0.062** | 1.42E-06 | **7.04E-04** | 0.974 | **1.22E-03** |
| | | w/o ES | 0.107 | 3.72E-06 | 1.23E-03 | 0.957 | 2.68E-03 |
| | | w/o BA | - | - | - | - | - |
| | RI | Full model | **0.090** | **2.75E-06** | **1.02E-03** | 0.955 | **8.74E-04** |
| | | w/o PF | 0.092 | 2.92E-06 | 1.04E-03 | **0.972** | 2.05E-02 |
| | | w/o DS | 0.092 | 2.85E-06 | 1.03E-03 | 0.944 | 9.10E-04 |
| | | w/o ES | 0.129 | 5.50E-06 | 1.45E-03 | 0.921 | 1.98E-03 |
| | | w/o BA | - | - | - | - | - |
| Kol. Flow | FI | Full model | 0.081 | 1.56E-01 | 2.66E-01 | **0.997** | **2.40E-01** |
| | | w/o PF | **0.077** | **1.41E-01** | **2.24E-01** | **0.997** | 7.99E-01 |
| | | w/o DS | 0.083 | 1.64E-01 | 2.72E-01 | **0.997** | 2.57E-01 |
| | | w/o ES | 0.340 | 2.65E+00 | 1.10E+00 | 0.950 | 3.86E-01 |
| | | w/o BA | 0.109 | 2.74E-01 | 3.24E-01 | 0.994 | 7.51E-01 |
| | RI | Full model | 0.253 | 1.48E+00 | 8.83E-01 | **0.975** | **5.65E-01** |
| | | w/o PF | **0.230** | **1.23E+00** | **7.16E-01** | **0.975** | 1.95E+00 |
| | | w/o DS | 0.259 | 1.54E+00 | 9.02E-01 | 0.974 | 6.30E-01 |
| | | w/o ES | 0.375 | 3.21E+00 | 1.33E+00 | 0.954 | 9.08E-01 |
| | | w/o BA | 0.276 | 1.74E+00 | 9.51E-01 | 0.971 | 1.12E+00 |
| RDGS | FI | Full model | 0.100 | 3.76E-03 | **2.46E-02** | **0.981** | **8.91E-08** |
| | | w/o PF | 0.107 | 4.25E-03 | 2.66E-02 | 0.980 | 1.58E-06 |
| | | w/o DS | **0.099** | **3.74E-03** | **2.46E-02** | **0.981** | 9.51E-08 |
| | | w/o ES | 0.214 | 1.52E-02 | 6.64E-02 | 0.929 | 9.00E-07 |
| | | w/o BA | 0.103 | 3.98E-03 | 2.52E-02 | 0.980 | 9.31E-08 |
| | RI | Full model | 0.193 | 1.35E-02 | 5.79E-02 | 0.941 | **1.69E-07** |
| | | w/o PF | 0.194 | 1.34E-02 | 5.76E-02 | 0.942 | 1.86E-06 |
| | | w/o DS | 0.193 | 1.33E-02 | 5.78E-02 | 0.942 | 1.90E-07 |
| | | w/o ES | **0.184** | **1.19E-02** | **5.62E-02** | **0.946** | 7.44E-07 |
| | | w/o BA | 0.196 | 1.39E-02 | 5.91E-02 | 0.940 | **1.69E-07** |

physics-aligned finetuning (PF) module significantly reduces the violation of physical laws, even in highly nonlinear and convection-dominant systems, which are typically challenging to optimize from scratch using a PINN-based objective (Krishnapriyan et al., 2021; Wang et al., 2022a). This success is largely due to a data-driven initial point provided by the pretraining stage.

Additionally, several modifications to the sampling process, as shown in Table 2, effectively enhance the final results. These include early-stop (ES), boundary-aware (BA), and deterministic sampling strategies (DS). Specifically, boundary-aware sampling effectively aligns with global effects introduced by boundary conditions, compelling the model to generate content that respects these conditions (see Appendix C.4). Interestingly, removing the Gaussian noise from PalSB's sampling path decreases physical dissatisfaction, and combining this with the early-stop strategy further improves performance. However, removing some modules may slightly enhance point-wise accuracy (nRMSE, MSE, MAE and correlation) while significantly deteriorating performance in terms of nER, indicating a severe breach of physical constraints. Further results and discussions on these ablations are available in Appendix C.

## 6    RELATED WORK

Data-driven reconstruction of physical field can date from the linear approximation theory such as POD (Berkooz et al., 1993; Everson & Sirovich, 1995; Borée, 2003; Li et al., 2023), Galerkin transforms (Noack & Eckelmann, 1994; Boisson & Dubrulle, 2011) and stochastic estimation (Adrian & Moin, 1988; Suzuki & Hasegawa, 2017), in which the performance on complicated systems are strongly limited by the linear assumption. Based on paired data. further attempts leverage the end-to-end modeling using neural networks, especially, CNN-based networks (Fukami et al., 2019; 2021a;b; Liu et al., 2020; Chai et al., 2020; Ren et al., 2023) and neural operator-based (Li et al.,

2020). Drawing inspirations from computer vision in image super-resolution (Ledig et al., 2017; Wang et al., 2018; 2021), adversarial loss and perceptual loss are introduced for field reconstruction (Yousif et al., 2021; Venkatesh et al., 2021; Li et al., 2023; Güemes et al., 2022). These methods either suffers from poor accuracy on challenging systems (Buzzicotti, 2023) nor struggling with the unstable adversarial training process (Brock et al., 2017; 2019; Miyato et al., 2018; Dhariwal & Nichol, 2021). More importantly, they can misalign with the corresponding physical constraints. Encoding of physical laws into field reconstruction are parially investigated in (Jacobsen et al., 2023; Shu et al., 2023; Bastek et al., 2024). In particular, instead of directly using PINN loss, PIDM (Shu et al., 2023) inject physical information through CDM in a classifier-free guidance manner. Notably, in many complex nonlinear systems like turbulence, optimizing PINN loss without a good initialization is difficult (Krishnapriyan et al., 2021; Wang et al., 2022a) while implicitly informing the physical information like PI-DDPM can easily make the model ignore this extra input.

The literature related to conditional generation of diffusion models and generation on constrained domain is included in Appendix D

## 7 DISCUSSION

In this work, we introduce the Physics-aligned Schrödinger Bridge (PalSB) framework, a novel approach for reconstructing physical fields from sparse measurements that effectively addresses the misalignment of physical laws often encountered with diffusion-based models. By employing a patch-based DSB for pretraining, our model achieves a robust initial weight configuration. This setup enhances the stability and efficacy of direct optimization when employing physics-informed losses, thereby preventing divergence. The PalSB framework is further augmented by a meticulously designed sampling process, enabling the accurate reconstruction of physical fields that adhere closely to physical constraints. Our approach is rigorously tested through practical tasks such as FI and RI, where PalSB demonstrates superior performance compared to baseline models across three different physical systems. These systems include challenging environments governed by highly nonlinear PDEs, such as 2D turbulence and reaction-diffusion systems.

The effectiveness of PalSB in these contexts underscores its potential for broad application in generating content that must conform to complex physical constraints within diffusion-based modeling frameworks. The successful implementation of PalSB not only paves the way for more accurate physical field reconstructions but also contributes to the evolving dialogue on integrating physical laws with advanced generative techniques in scientific computing.

## 8 LIMITATIONS

While the PalSB framework effectively aligns generated physical fields with their governing equations, it is important to acknowledge certain limitations. Firstly, the encoding of constraints within PalSB is implemented in a soft manner, implying that some residual discrepancies from the exact equations are inevitable. This soft constraint approach, while facilitating greater flexibility and computational feasibility, does not fully eliminate equation residuals. On the other hand, directly generating complex physical systems, such as those involving turbulence, on the exact solution manifold of the governing equations would be ideal. However, this remains a significant challenge due to the intricate dynamics and high nonlinearity inherent in such systems. Achieving this level of precision and adherence to the governing equations in a generative model is an area of ongoing research and development. Moreover, this study is constrained by computational resources, which has limited our validation to two-dimensional (2D) examples. The extension of our framework to three-dimensional (3D) cases, which are more representative of real-world scenarios, has not yet been tested but is a critical step for future work. This expansion to 3D will enable a more comprehensive assessment of the model's capabilities and applicability across a broader range of scientific and engineering problems.

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

## A    IMPLEMENTATION DETAILS

### A.1    PALSB

The algorithms for pretraining, finetuning and sampling of PalSB are included in Algs. 1, 2 and 3, respectively. The interpolation function $\mathcal{I}(\cdot)$, aiming to interpolate from the input space of $\mathbf{y}$ to the output space of $\mathbf{x}_0$, is chosen to be bi-cubic for FI task while Voronoi tessellation (Fukami et al., 2021b) for RI task.

Compared to training a denoising network that starts from pure noise (such DDPM), the mapping relations for DSB (which start directly from a low-fidelity sample) is easier to learn. Therefore, the scalable neural network used to predict the high-fidelity field can be a simplified U-Net equipped with residual blocks and channel-wise linear attention blocks as suggested in (Luo et al., 2023), removing the group normalization and conventional attention blocks used in DDPM (Ho et al., 2020; Song et al., 2021). The hyperparameters of the network architecture are listed in Table 6.

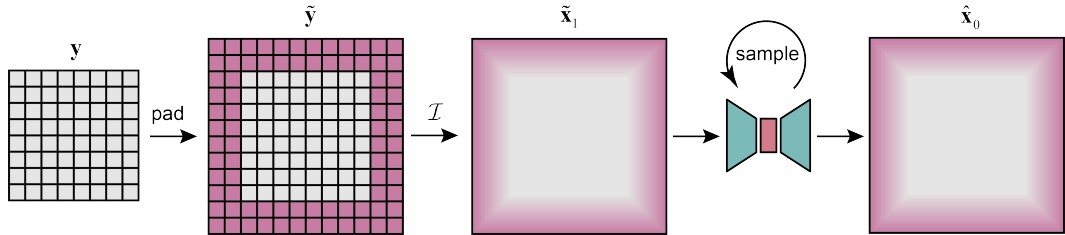

Figure 4: Boundary-aware sampling strategy

The schematic of BA strategy is demonstrated in Fig. 4, where the input low-fidelity data is first padded according to the boundary condition, and then interpolated to match the dimension of high-resolution output. Subsequently, the sampling process is performed on the padded input, after which the padded part of the generated field is trimmed to obtain the final output. Another trick of ES strategy is illustrated in Fig. 5. Under the same NFEs, instead of using a larger step size that covers the full sampling path, we stop the sampling procedure at the early stage with a small step size, regarding the model output from intermediate sample as the final prediction.

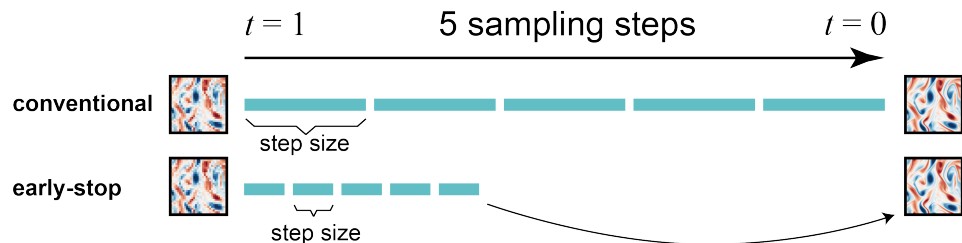

Figure 5: Early-stop sampling strategy

### A.2    BASELINES

**For the E2E method**, we use the FNO (Li et al., 2020), which is a powerful tool for learning the mapping between function space, to fit the direct mapping from low-fidelity field to the high-fidelity one. Notably, the low-fidelity input is also interpolated from the original measurements to keep the same with our method. Since FNO can inherently capture the global interactions within spatial area, we train the model on full fields instead of randomly cropped patches. To make fair comparison, the E2E model is also finetuned using the physics-informed objective as we used for finetuning the PalSB (Eq. 7). The hyperparameters of FNO we used are listed in Table 7.

**For the PIDM method**, we follow all the settings of in the original paper (Shu et al., 2023) that conditioned on equation residual information. Specifically, we use the pretrained weights for the case of Kolmogorov flow published in (Shu et al., 2023), while training the models for cylinder flow and

---

**Algorithm 1** Pretraining of PalSB

---

**Require:** Training dataset $\{\mathbf{x}_0^{(i)}, \mathbf{y}^{(i)}\}_{i=0}^N$, initialized neural network $f_\theta$ with parameter group $\theta$, diffusion schedule $\beta(t)$, interpolation function $\mathcal{I}(\cdot)$

1: **while** not convergence **do**
2:   Sample $i \sim U(\{1, 2, ..., N\})$         ▷ Get sample from training dataset
3:   $\tilde{\mathbf{x}}_0^{(i)}, \tilde{\mathbf{y}}^{(i)} \leftarrow \text{crop}(\mathbf{x}_0^{(i)}), \text{crop}(\mathbf{y}^{(i)})$       ▷ Apply random cropping
4:   $\tilde{\mathbf{x}}_1^{(i)} \leftarrow \mathcal{I}(\tilde{\mathbf{y}}^{(i)})$             ▷ Interpolate
5:   Sample $t \sim U(0, 1)$
6:   $\sigma_t^2 \leftarrow \int_0^t \beta(\tau)d\tau, \bar{\sigma}_t^2 \leftarrow \int_t^1 \beta(\tau)d\tau$
7:   $C_1 \leftarrow \frac{\bar{\sigma}_t^2}{\bar{\sigma}_t^2 + \sigma_t^2}, C_2 \leftarrow 1 - C_1$
8:   $\tilde{\mathbf{x}}_t^{(i)} \sim \mathcal{N}\left(C_1 \tilde{\mathbf{x}}_0^{(i)} + C_2 \tilde{\mathbf{x}}_1^{(i)}, C_1 \sigma_t^2 \mathbf{I}\right)$
9:   $\hat{\mathbf{x}}_0^{(i)} \leftarrow f_\theta(\tilde{\mathbf{x}}_t^{(i)}, \tilde{\mathbf{x}}_1^{(i)}, t)$        ▷ Get prediction at step $t$
10:   $\mathcal{J}(\theta) \leftarrow \|\hat{\mathbf{x}}_0^{(i)} - \tilde{\mathbf{x}}_0^{(i)}\|$      ▷ Calculate the loss accroding to eq. 6
11:   Optimize $\mathcal{J}(\theta)$          ▷ Optimize the neural network
12: **end while**
13: $\theta^* \leftarrow \theta$

---

**Algorithm 2** Physics-aligned finetuning of PalSB

---

**Require:** Training dataset $\{\mathbf{x}_0^{(i)}, \mathbf{y}^{(i)}\}_{i=0}^N$, initialized neural network $f_\theta$ with parameter group $\theta$, pretrained weights $\theta^*$, physical constraints $\mathcal{F}(\cdot)$, diffusion schedule $\beta(t)$, number of sampling steps $T$, number of backpropagation step $B$, weights $\alpha, \beta$

1: $\theta \leftarrow \theta^*$           ▷ Start from the pretrained weights
2: Assign $0 < t_T < t_{T-1} < ... < t_1 = 1$      ▷ Time scheduling
3: **while** not convergence **do**
4:   Sample $\mathbf{x}_0, \mathbf{y} \sim U(\{\mathbf{x}_0^{(i)}, \mathbf{y}^{(i)}\}_{i=0}^N)$    ▷ Get sample from training dataset
5:   $\mathbf{x}_1 \leftarrow \mathcal{I}(\mathbf{y})$           ▷ Interpolate
6:   $j \leftarrow 1$
7:   $\mathbf{x} \leftarrow \mathbf{x}_1$
8:   **while** $j < T$ **do**           ▷ Sampling loop
9:    **if** $j < T - B$ **then**
10:     $\mathbf{x} \leftarrow \text{sg}(\mathbf{x})$      ▷ Truncate the gradient to save memory
11:    **end if**
12:    $\hat{\mathbf{x}}_0 \leftarrow f_\theta(\mathbf{x}, \mathbf{x}_1, t_j)$
13:    $\sigma_{t_j}^2 \leftarrow \int_0^{t_j} \beta(\tau)d\tau; \bar{\sigma}_{t_j}^2 \leftarrow \int_{t_j}^1 \beta(\tau)d\tau$
14:    $C_1 \leftarrow \frac{\bar{\sigma}_{t_j}^2}{\bar{\sigma}_{t_j}^2 + \sigma_{t_j}^2}; C_2 \leftarrow 1 - C_1$
15:    $\mathbf{x} \sim \mathcal{N}\left(C_1 \hat{\mathbf{x}}_0 + C_2 \mathbf{x}_1, C_1 \sigma_{t_j}^2 \mathbf{I}\right)$
16:    $j \leftarrow j + 1$
17:   **end while**
18:   $\mathcal{J}(\theta) \leftarrow \alpha\|\hat{\mathbf{x}}_0 - \mathbf{x}_0\| + \beta\|\mathcal{F}(\hat{\mathbf{x}}_0)\|$    ▷ Calculate the loss according to eq. 7
19:   Optimize $\mathcal{J}(\theta)$
20: **end while**

---

---

**Algorithm 3** Sampling process of PalSB

---

**Require:** Measurements $\mathbf{y}$, trained neural network $f_\theta$, diffusion schedule $\beta(t)$, number of sampling steps $T$, boundary padding function $\mathrm{pad}(\cdot)$ and the corresponding boundary trimming function $\mathrm{trim}(\cdot)$

1: Assign $0 < t_T < t_{T-1} < ... < t_1 = 1$            ▷ Time scheduling
2: $\mathbf{y} \leftarrow \mathrm{pad}(\mathbf{y})$         ▷ Pad the field according to the boundary condition
3: $\mathbf{x}_1 \leftarrow \mathcal{I}(\mathbf{y})$           ▷ Interpolate
4: $\mathbf{x} \leftarrow \mathbf{x}_1$
5: $j \leftarrow 1$
6: **while** $j < T$ **do**          ▷ Sampling loop with early-stop strategy
7:      $\hat{\mathbf{x}}_0 \leftarrow f_\theta(\mathbf{x}, \mathbf{x}_1, t_j)$
8:      $\sigma_{t_j}^2 \leftarrow \int_0^{t_j} \beta(\tau) d\tau; \bar{\sigma}_{t_j}^2 \leftarrow \int_{t_j}^1 \beta(\tau) d\tau$
9:      $C_1 \leftarrow \frac{\bar{\sigma}_{t_j}^2}{\bar{\sigma}_{t_j}^2 + \sigma_{t_j}^2}; C_2 \leftarrow 1 - C_1$
10:      $\mathbf{x} \leftarrow C_1 \hat{\mathbf{x}}_0 + C_2 \mathbf{x}_1$          ▷ Deterministic sampling path
11:      $j \leftarrow j + 1$
12: **end while**
13: $\hat{\mathbf{x}}_0 \leftarrow \mathrm{trim}(\hat{\mathbf{x}}_0)$          ▷ Trim the padded boundary grids

---

reaction-diffusion system from scratch. Note that the DDPM-based method which start from pure Gaussian noise cannot easily scale up to larger domain (Hoogeboom et al., 2023). Therefore, we also train the PIDM on full fields instead of randomly cropped patches. All the neural networks for PIDM share the same architecture, where the hyperparameters are listed in the original paper. Based on SDEdit and classifier-free guidance, the sampling process of PIDM in our experiments follows the original paper that uses the same parameters as suggested in its Github repository (Shu et al., 2023).

### A.3 DATASETS DESCRIPTION

**Cylinder flow measured by PIV.** Due to the limitation of PIV experiment, the pressure field is not accessible, which means the governing equations cannot be fully characterized as physical constraints. Here, we only consider the constraints of continuity (which refer to the mass conservatin of the fluid) as following

$$\nabla \cdot \mathbf{u} = 0 \tag{9}$$

where $\nabla \cdot$ is the divergence operator and $\mathbf{u}$ is the 2D velocity vector field.

**2D forced turbulence.** The Kolmogorov flow follows the vorticity equation derived from Navier-Stokes equations written as

$$\frac{\partial \omega}{\partial t} + \mathbf{u} \cdot \nabla \omega - \frac{1}{Re} \nabla^2 \omega = \mathbf{f} \tag{10}$$

where $\mathbf{u} \equiv [u, v]$ is the velocity vector field, $\omega \equiv \partial v/\partial x - \partial u/\partial y$ is the vorticity field, Reynolds number $Re$ is a constant scalar and $\mathbf{f}$ is the external force defined in (Shu et al., 2023). Assuming the periodic boundary condition, the velocity in this equation can be easily determined by solving a Poisson equation in Fourier space. Therefore, the equation residual can be calculated given a sample of vorticity field.

**Reaction-diffusion system.** The governing equation of RDGS is written as

$$\frac{\partial u}{\partial t} = \mu_u \Delta u - uv^2 + F(1 - u)$$
$$\frac{\partial v}{\partial t} = \mu_v \Delta v + uv^2 - (F + \kappa)v \tag{11}$$

where $u$ and $v$ are the concentration variables, $\mu_u$ and $\mu_v$ are the diffusion coefficients, $F$ and $\kappa$ are the parameters that control the reaction source terms, and $\Delta$ denotes the Laplacian operator. We use finite difference method to calculate the derivatives in the equations.

## A.4 RUNTIME ENVIRONMENT

All the experiments conducted in this paper are running on a single GPU of Nvidia GeForce RTX 3090 with Intel(R) Xeon(R) Gold 6226R CPU. The platform is Ubuntu 20.04.3 LTS operation system with Python 3.9 environment. We list the parameters for reproducing our experiments in Tables 8 and 9, including the training, finetuning and sampling process.

## B FURTHER RESULTS

Table 3: Performance comparison of models on NS3D FI and NS3D RI cases with noise-free and noisy data.

| Case | Model | nRMSE ↓ | MSE ↓ | L1 Error ↓ | Correlation ↑ | nER ↓ |
|------|-------|---------|-------|-----------|---------------|-------|
| NS3D FI (noise-free) | PalSB | **0.0928** | **0.0043** | **0.0457** | **0.9954** | **3.8502** |
| | interp. | 0.3263 | 0.0539 | 0.1663 | 0.9415 | 14.0915 |
| NS3D FI (5% noise) | PalSB | **0.0948** | **0.0045** | **0.0468** | **0.9953** | **3.8700** |
| | interp. | 0.3267 | 0.0540 | 0.1665 | 0.9414 | 14.2528 |
| NS3D RI (noise-free) | PalSB | **0.1846** | **0.0168** | **0.0929** | **0.9820** | **2.4647** |
| | interp. | 0.3645 | 0.0669 | 0.1758 | 0.9299 | 625.9368 |
| NS3D RI (5% noise) | PalSB | **0.1850** | **0.0168** | **0.0931** | **0.9820** | **2.4701** |
| | interp. | 0.3654 | 0.0672 | 0.1766 | 0.9296 | 629.2205 |

### B.1 EXTENSION ON 3D DATASET

To validate the capability of our methodology beyond 2D cases, we further test the PalSB on a complex 3D system for both FI and RI tasks. The 3D dynamical system is describe by the compressible Navier-Stokes equations, which contains the following continuity equation and momentum equation

$$
\begin{aligned}
\partial_t \rho + \nabla \cdot (\rho \mathbf{v}) &= 0 \\
\rho \left( \partial_t \mathbf{v} + \mathbf{v} \cdot \nabla \mathbf{v} \right) &= -\nabla p + \eta \Delta \mathbf{v} + (\zeta + \eta/3) \nabla (\nabla \cdot \mathbf{v})
\end{aligned}
\tag{12}
$$

where $\mathbf{v}$ is the 3D velocity vector, $\rho$ is the density, $p$ is the pressure, and $\zeta$ and $\eta$ are bulk and shear viscosity, respectively. while challenging due to the highly nonlinear nature of the governing equations, this problem is significant in many applications like aerodynamics.

The training and testing dataset is drawn from PDEBench Takamoto et al. (2022) with $\eta = \zeta = 1e-8$, periodic boudaries and turbulent initial conditions, which contains 600 trajectories of $\mathbf{v}$, $\rho$ and $p$. Each trajectory contains 21 snapshots spatially discretized on a $64 \times 64 \times 64$ grid. We train our model on the first 90% of the trajectories and test on the rest. We perform 4x super-resolution for FI task and 1% observation reconstruction for RI task. As illustrated in Figs. 20,21 and Table 3, PalSB can still make accurate predictions in 3D configurations, representing the power of PalSB in modeling a wide range of physical systems.

### B.2 NOISY OBSERVATIONS

In order to assess the robustness of the models, we add Gaussian noise with 5% of the standard deviation of the corresponding data into the input low-fidelity field. As shown in Table 4, PalSB maintains similar performance as in noise-free cases. Particularly, in the two tasks of reaction-diffusion system where there is large area that is not diffused, PalSB recovers these areas even perturbed with noise (Fig. 18 and 19).

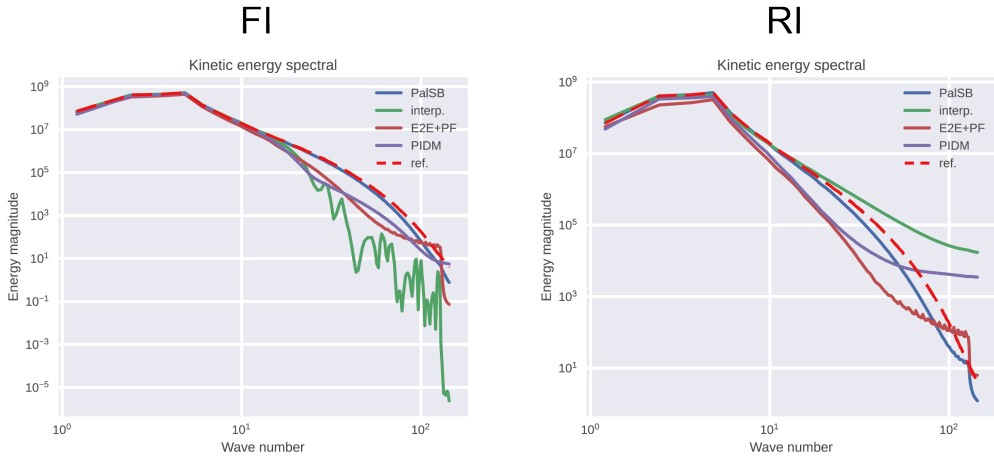

Figure 6: Statistical comparison of the methods on Kolmogorov flow.

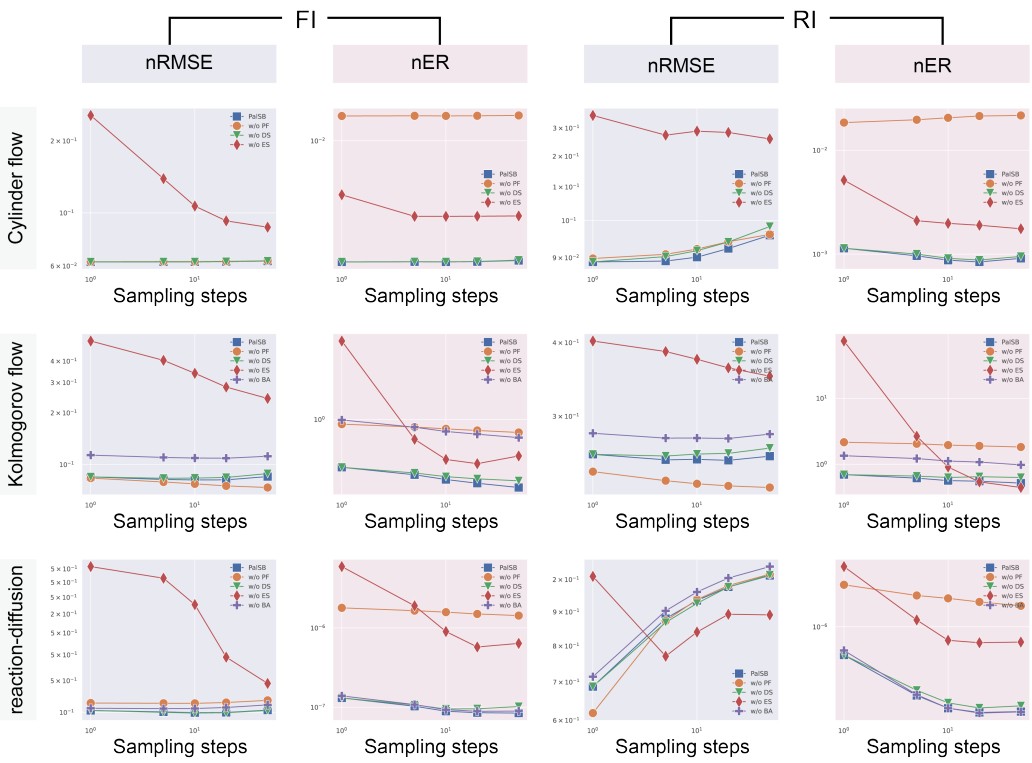

Figure 7: Ablations on sampling steps.

### B.3 STATISTICS OF THE KOLMOGOROV FLOW

Kinetic energy spectral (KES) of the turbulent flow (Boffetta & Ecke, 2012), vital statistics of the energy distribution on varying spatial frequency components, is calculated for the fields predicted by different methods as shown in Fig. 6, where the red dot-line represents the KES from the high-fidelity simulation data. In both tasks, the turbulent fields generated by PalSB can better capture the kinetic energy distribution in spectral space than other methods, indicating the efficacy of PalSB in remembering the data statistics.

## C    ABLATIONS

### C.1    NUMBER OF SAMPLING STEPS

As shown in Figs. 7 and 11, though the increase of sampling steps does not necessarily facilitate the accuracy and physical satisfaction, in most cases, more sampling steps can reduce the nER. These phenomena can be first attributed to the restricted sampling steps for finetuning that the model gets overfitted around the configured number of sampling steps. Besides, as investigated in diffusion-based image restoration studies (Luo et al., 2023), the multi-step generation through the diffusion path represent particular capability in generating contents that look like the samples in training dataset, which, in other word, can keep statistical consistency with the target distribution of the dataset while may diverge on the point-wise accuracy. Consequently, we choose 10 steps to sample the field that makes a proper trade-off between performance and time consumption.

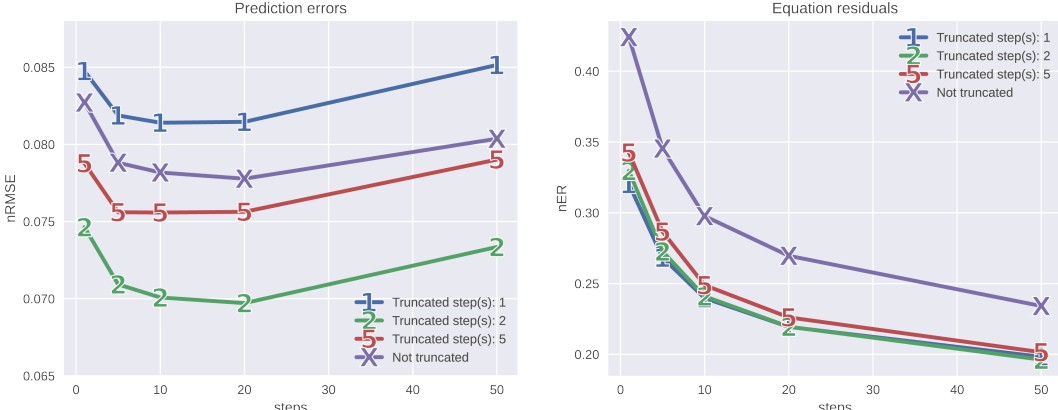

Figure 8: Ablations on the number of truncated steps in finetuning, which is tested on FI task for Kolmogorov flow.

### C.2    NUMBER OF TRUNCATED STEPS IN FINETUNING

In the finetuning stage, the backpropagation along the sampling path is truncated to save the computational consumption. The number of truncated steps in finetuning (i.e., $B$ in Alg. 2) can affect the performance of PalSB as seen in Fig. 8. In addition to the higher computational costs, we find that a larger number of backpropagation steps does not necessarily enhance the results. Excessive steps of backpropagation might lead to difficulty for the model's convergence to the global optimum within a given number of iterations.

### C.3    DIFFUSION SCHRÖDINGER BRIDGE

Based on the observation mentioned above that he increase of sampling steps does not necessarily facilitate the accuracy of the model prediction, we further train the same neural network without using diffusion path, which is, in fact, an E2E model that directly optimizes the regression loss. As demonstrated in Table 5, we find that even such model can achieve a higher accuracy than the model trained by diffusion-like loss, it is still not convincing in characterizing the statistical features of physical field and shows higher violation against the physics. Moreover, such deterministic model is unable to evaluate the uncertainty.

### C.4    BOUNDARY-AWARE SAMPLING

Considering that the scalability of the neural network used for PalSB results in obstacles for capturing global patterns that introduced by boundary conditions, this strategy is critical for PalSB to enforce the boundary conditions as illustrated in Fig. 3. Here, the periodic boundary condition is well aligned through the sampling on the padded field, which enforce the model to be aware of the global information in a local manner. As shown in Fig. 3, padding is applied to the low-fidelity observation

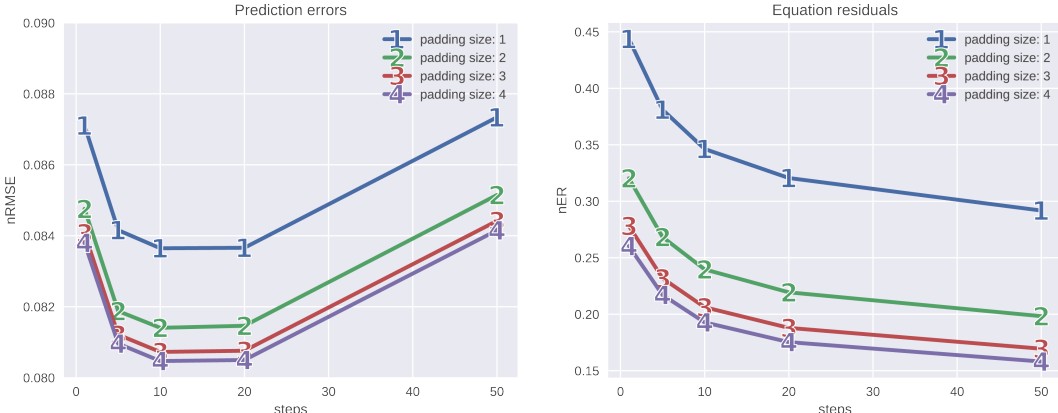

Figure 9: Ablations on the size of padding in sampling, which is tested on FI task for Kolmogorov flow.

**y** before interpolation to the original grid. Consequently, the choice of padding size for FI and RI tasks depends on the size of the observation. Specifically, the padding size along each direction is set to 2 for the FI task and 16 for the RI task, respectively. This ensures consistency in the size of the interpolated data samples in each case.

We further examine the impact of padding size on the final performance in the Kolmogorov flow case. As illustrated in Fig. 9, increasing the padding size improves the final performance. However, this improvement comes with additional computational costs, as the neural network must process a larger input. Therefore, a trade-off between performance and computational cost should be carefully considered when employing boundary-aware sampling.

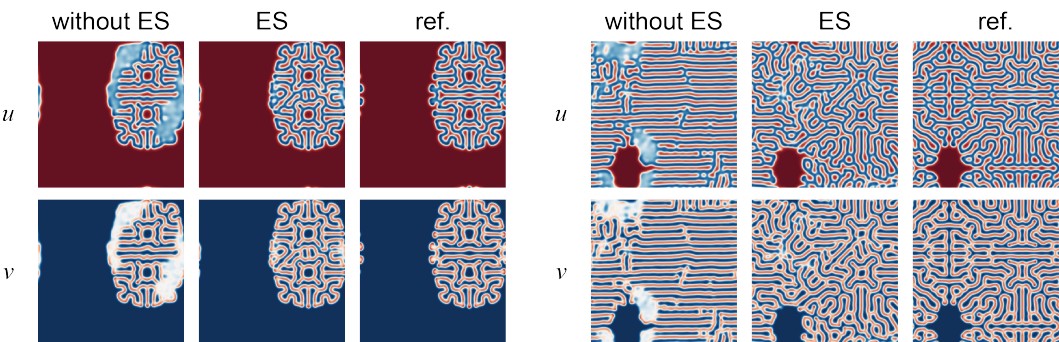

Figure 10: Ablations on early-stop sampling strategy. This reaction-diffusion case on RI task indicates that a lower nRMSE do not mean physically reasonable.

### C.5 EARLY-STOP SAMPLING STRATEGY

Under the same number of function evaluation (NFE), we find that sampling with a small time step and then stopping at the early stage is a better strategy than increasing the step size as implemented in existing work (Fig. 7 and Fig. 11). Smaller step size can possibly reduce the discretization error for the first few steps on sampling path to more accurately simulate the sample from the posterior distribution at intermediate step, yielding significant improvements in most experiments when the NFEs are relatively low (e.g., less than 50 NFEs). Although the nRMSE on reaction-diffusion system shows that the model without early-stop sampling is better, we find its failure in generated spatial patterns as illustrated in Fig. 10, indicating the bias for only using the point-wise accuracy as the evaluation metric.

## C.6 DETERMINISTIC SAMPLING

Removing all the Gaussian noise in sampling path leads to a deterministic sampling process that is the same as OT-ODE in (Liu et al., 2023b). To our surprise, such procedure shows enhancement on nER in our experiments, even the model is not trained with deterministic DSB (i.e., not trained with OT-ODE). We suppose that this phenomenon is related to the symmetric noise schedule of DSB, which is nearly noise-free at the two ends of the sampling path, making it possible for the neural network to generalize to noise-free input at the early stage of the sampling process.

Table 4: Performance comparison of models across different cases and tasks on noisy observations (5% Gaussian noise). The metrics with blue color evaluate the errors with the reference data, while the metric with red color evaluates the physics compliance.

| Case | Task | Model | nRMSE ↓ | MSE ↓ | MAE ↓ | Correlation ↑ | nER ↓ |
|------|------|-------|---------|-------|-------|---------------|-------|
| Cy. Flow | FI | Interp. | 0.276 | 2.33E-05 | 3.13E-03 | 0.842 | 6.64E-02 |
| | | E2E+PF | 0.094 | 3.42E-06 | 1.09E-03 | 0.972 | 8.35E-03 |
| | | PIDM | 0.262 | 2.14E-05 | 3.05E-03 | 0.768 | 1.59E-02 |
| | | SB | **0.063** | **1.37E-06** | 7.22E-04 | **0.994** | 1.54E-02 |
| | | PalSB | **0.063** | 1.43E-06 | **7.10E-04** | 0.975 | **1.23E-03** |
| | RI | Interp. | 0.303 | 2.85E-05 | 3.19E-03 | 0.890 | 1.73E+00 |
| | | E2E+PF | 0.100 | 3.82E-06 | 1.17E-03 | 0.909 | 9.59E-03 |
| | | PIDM | 0.121 | 4.98E-06 | 1.45E-03 | 0.899 | 4.59E-02 |
| | | SB | 0.093 | 2.95E-06 | 1.05E-03 | **0.971** | 2.04E-02 |
| | | PalSB | **0.091** | **2.79E-06** | **1.03E-03** | 0.953 | **8.83E-04** |
| Kol. Flow | FI | Interp. | 0.540 | 6.62E-01 | 1.88E+00 | 0.856 | 1.52E-01 |
| | | E2E+PF | 0.288 | 1.92E-01 | 1.04E+00 | 0.959 | 3.93E-01 |
| | | PIDM | 0.524 | 6.23E-01 | 1.83E+00 | 0.859 | 3.84E-01 |
| | | SB | **0.085** | **1.68E-01** | **2.61E-01** | **0.996** | 8.54E-02 |
| | | PalSB | 0.088 | 1.79E-01 | 2.91E-01 | **0.996** | **2.72E-02** |
| | RI | Interp. | 0.584 | 7.75E-01 | 1.86E+00 | 0.840 | 6.27E+02 |
| | | E2E+PF | 0.466 | 5.00E-01 | 1.72E+00 | 0.908 | 2.70E+00 |
| | | PIDM | 0.382 | 3.33E-01 | 1.34E+00 | 0.934 | 2.79E+01 |
| | | SB | **0.233** | **1.26E-01** | **7.33E-01** | **0.972** | 1.98E+00 |
| | | PalSB | 0.255 | 1.50E+00 | 8.90E-01 | **0.972** | **5.73E-01** |
| RDGS | FI | Interp. | 0.284 | 2.61E-02 | 1.06E-01 | 0.861 | 4.63E-04 |
| | | E2E+PF | 0.261 | 2.18E-02 | 1.03E-01 | 0.883 | 8.18E-05 |
| | | PIDM | 0.268 | 2.31E-02 | 1.06E-01 | 0.870 | 1.30E-03 |
| | | SB | 0.107 | 4.27E-03 | 2.67E-02 | 0.980 | 1.59E-06 |
| | | PalSB | **0.100** | **3.77E-03** | **2.47E-02** | **0.981** | **9.13E-08** |
| | RI | Interp. | 0.294 | 2.81E-02 | 1.04E-01 | 0.851 | 3.02E-02 |
| | | E2E+PF | 0.394 | 5.02E-02 | 1.63E-01 | 0.701 | 5.39E-05 |
| | | PIDM | 0.217 | 1.55E-02 | 9.20E-02 | 0.918 | 3.92E-03 |
| | | SB | **0.194** | **1.34E-02** | **5.78E-02** | **0.941** | 1.87E-06 |
| | | PalSB | **0.194** | 1.35E-02 | 5.81E-02 | 0.939 | **1.74E-07** |

# D FURTHER DISCUSSION ON RELATED WORK

## D.1 CONDITIONAL GENERATION WITH DIFFUSION MODELS

Diffusion models (Song & Ermon, 2019; Ho et al., 2020; Song et al., 2021) are widely applied to the generation of diverse contents, especially image (Dhariwal & Nichol, 2021; Rombach et al., 2022; Saharia et al., 2022), video (Ho et al., 2022b;a; Blattmann et al., 2023) and molecular (Xu et al., 2022; Watson et al., 2023; Igashov et al., 2024). Particularly, conditional generation using diffusion model based on given information is highly regarded in practical scenes such as image/video restoration (Dhariwal & Nichol, 2021; Ho & Salimans, 2022; Meng et al., 2022; Song et al., 2022; Mardani et al., 2023; Chung et al., 2023; Song et al., 2021; Wang et al., 2022b; Yue et al., 2023; Luo et al., 2023; Zhang et al., 2023; Liu et al., 2023b), molecular generation (Shi et al., 2021; Xu et al., 2022; Watson et al., 2023; Igashov et al., 2024; Vecchio et al., 2024; Didi et al., 2024) and dynamic forecasting (Gao et al., 2023; Yoon et al., 2023; Cachay et al., 2023), basically categorized into two classes that based on unconditional diffusion model (UDM) and conditional diffusion model

Table 5: Ablation on DSB training

| Case | Task | Model | nRMSE ↓ | MSE ↓ | MAE ↓ | Correlation ↑ | nER ↓ |
|------|------|-------|---------|-------|-------|---------------|-------|
| Cy. Flow | FI | w/o DSB | 0.063 | 1.41E-06 | 7.18E-04 | 0.973 | 1.38E-03 |
|  |  | w/ DSB (10 steps) | 0.062 | 1.42E-06 | 7.04E-04 | 0.974 | 1.22E-03 |
|  | RI | w/o DSB | 0.081 | 2.35E-06 | 8.98E-04 | 0.967 | 1.44E-03 |
|  |  | w/ DSB (10 steps) | 0.090 | 2.75E-06 | 1.02E-03 | 0.955 | 8.74E-04 |
| Kol. Flow | FI | w/o DSB | 0.075 | 1.34E-01 | 2.30E-01 | 0.998 | 3.34E-01 |
|  |  | w/ DSB (10 steps) | 0.081 | 1.56E-01 | 2.66E-01 | 0.997 | 2.40E-01 |
|  | RI | w/o DSB | 0.223 | 1.14E+00 | 7.98E-01 | 0.983 | 1.01E+00 |
|  |  | w/ DSB (10 steps) | 0.253 | 1.48E+00 | 8.83E-01 | 0.975 | 5.65E-01 |
| RDGS | FI | w/o DSB | 0.092 | 3.23E-03 | 2.21E-02 | 0.984 | 6.38E-08 |
|  |  | w/ DSB (10 steps) | 0.100 | 3.76E-03 | 2.46E-02 | 0.981 | 8.91E-08 |
|  | RI | w/o DSB | 0.140 | 6.86E-03 | 4.18E-02 | 0.967 | 1.06E-06 |
|  |  | w/ DSB (10 steps) | 0.193 | 1.35E-02 | 5.79E-02 | 0.941 | 1.69E-07 |

(CDM) respectively. Utilizing a pretrained UDM as *a priori* distribution, methods such as (Ho & Salimans, 2022; Chung et al., 2023; Song et al., 2022; Wang et al., 2022b; Mardani et al., 2023) manipulate the generating path towards the required condition, which is usually performed through the gradient-based correction. By partially noising the conditions along the forward diffusion path and then running the reversed process, SDEdit (Meng et al., 2022) can approximately sample from the target conditional distribution. ControlNet is another popular tool appended to a pretrained UDM, injecting conditional information through finetuning an additional feature adjusting network on paired data. Classifier-free guidance (Ho & Salimans, 2022) uses a weighted combination of UDM and CDM to guide the sampling path. Instead of training a UDM that requires intensive data and computational resources, CDM directly add the conditional information as an additional input of the model. As a special case of CDM, instead of starting from Gaussian noise, DSB (De Bortoli et al., 2021; Bunne et al., 2023; Liu et al., 2023b; Tong et al., 2024; Yue et al., 2023) directly learns the probabilistic path from the conditions to the targets, which is much more fast-to-train and data-efficient for specific task. Despite the empirically proved efficacy on image/audio tasks, the modeling of physical field using DSB is not well studied, which is addressed in this work.

### D.2 Generation on constrained domain

Diffusion model is adept to characterize the statistics of training data yet does not guarantee the satisfaction of constraints for the generated contents. There are efforts to apply simple constraints on diffusion models. Specifically, pixel-wise thresholding during generation (Ho et al., 2020; Saharia et al., 2022; Lu et al., 2023) is a simple while effective trick to fulfill the value range of an image. Similarly, mirror diffusion model (Liu et al., 2023a; Lou & Ermon, 2023) leverages the mirror mapping to restrict the generation not exceeding a given convex set. Generation with equaivariance (Hoogeboom et al., 2022; Igashov et al., 2024) is another class of methods that are ubiqitous in molecular related problems. Projected generative diffusion model (Christopher et al., 2024) further extended the constraints to more complicated domain on the support of ΠGDM (Song et al., 2022). However, these methods can only work on tractable domain of constraints, failing when the sample is embedded on a complicated and intractable manifold such as the physical field that derived from the solution of highly nonlinear PDEs.

### E   Code and data availability

The code for PalSB is submitted as the supplementary material. The data and trained checkpoints can be downloaded at the Google Drive after the paper's acceptance.

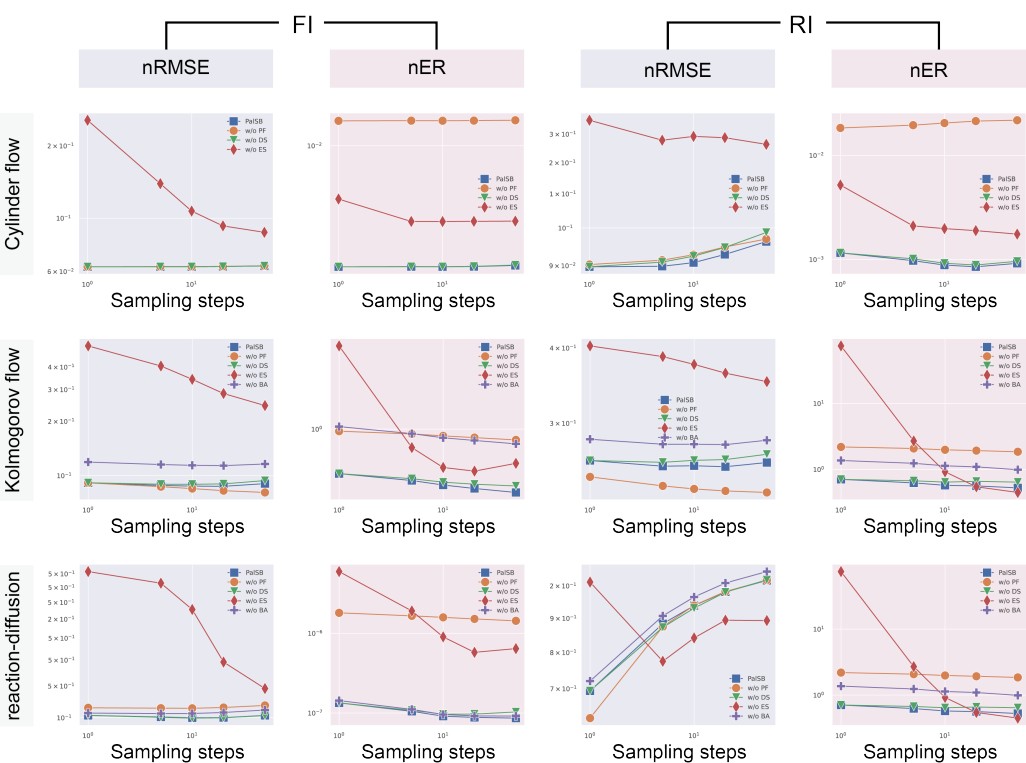

Figure 11: Ablations on sampling steps (5% noise).

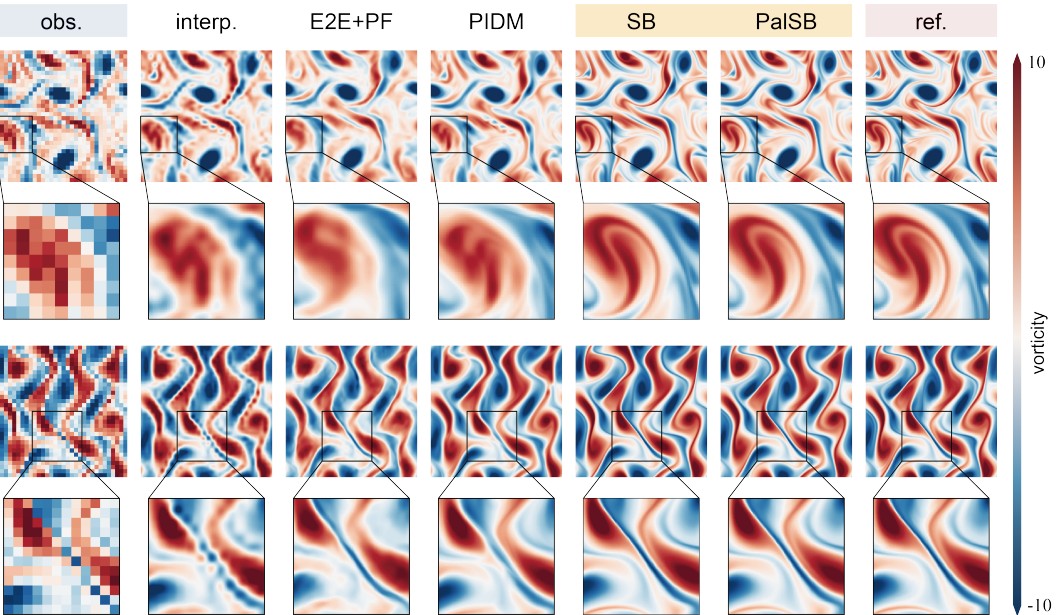

Figure 12: Additional results of Kolmogorov flow on FI task (8x super-resolution, noise-free).

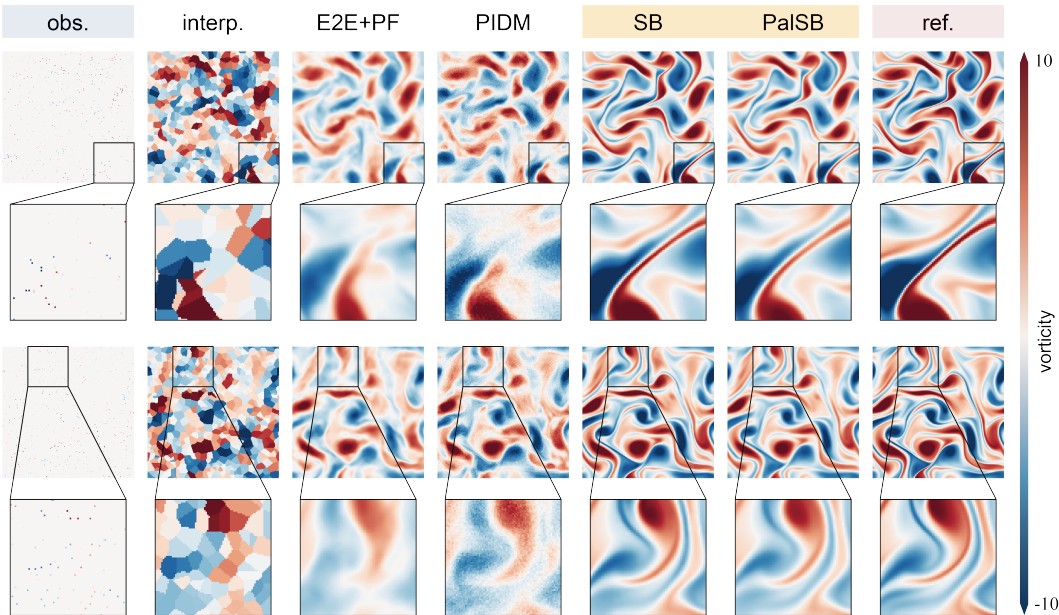

Figure 13: Additional results of Kolmogorov flow on RI task (99% masked, noise-free).

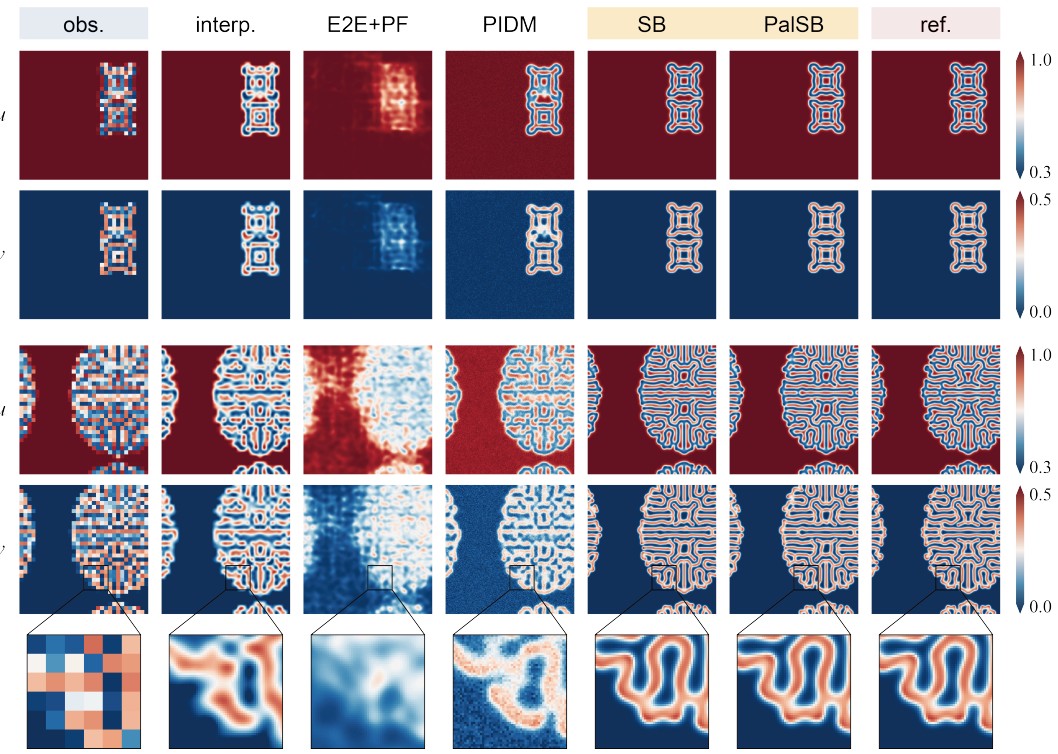

Figure 14: Additional results of GS-RD on FI task using noise-free observations (8x super-resolution, 0% Gaussian noise).

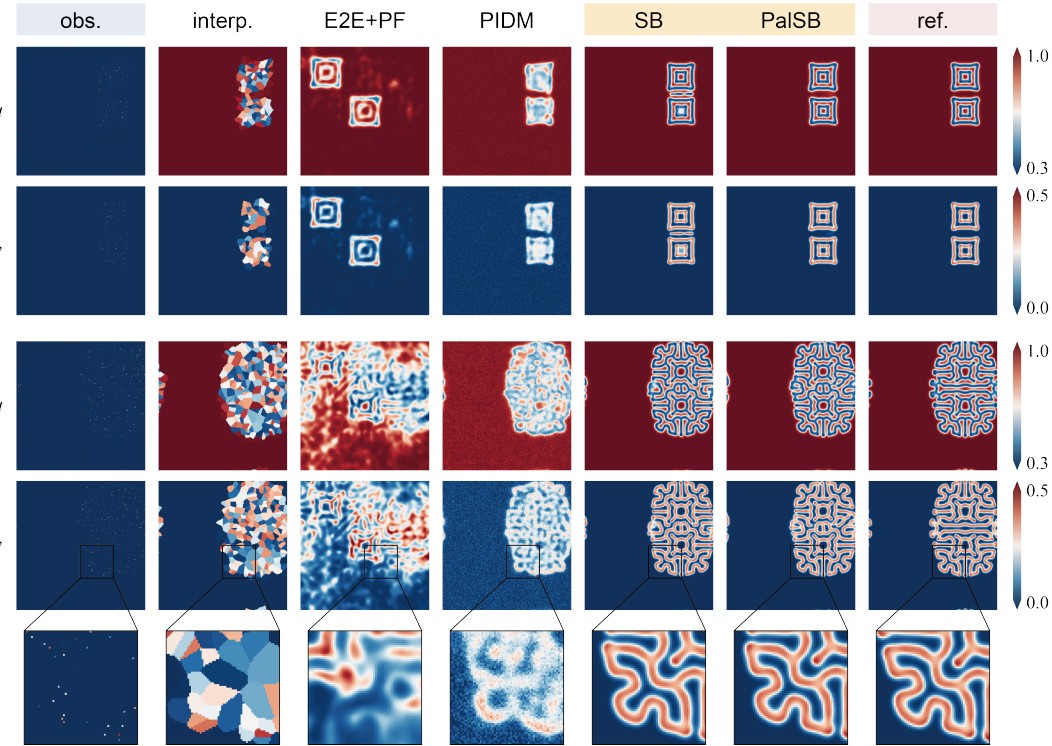

Figure 15: Additional results of GS-RD on RI task using noise-free observations (99% masked, noise-free).

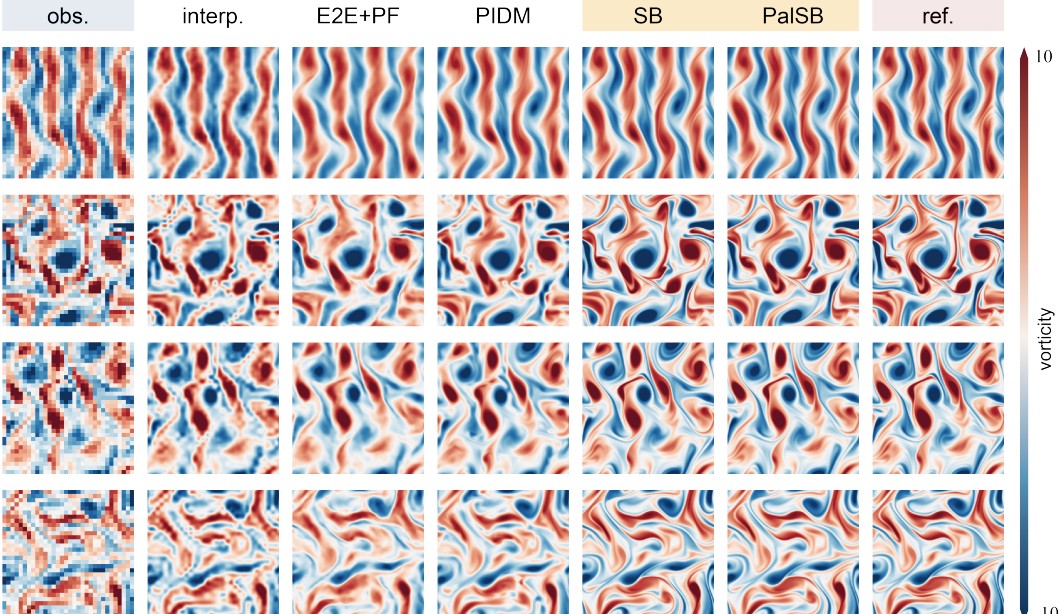

Figure 16: Additional results of Kolmogorov flow on FI task using noisy observations (8x super-resolution, 5% Gaussian noise).

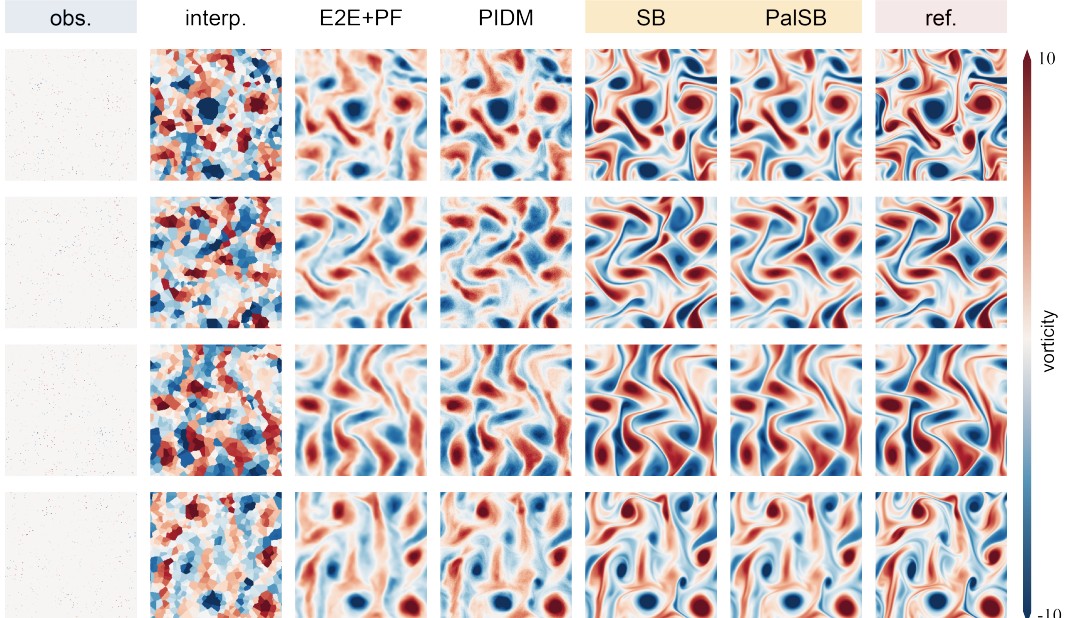

Figure 17: Additional results of Kolmogorov flow on RI task using noisy observations (99% masked, 5% Gaussian noise).

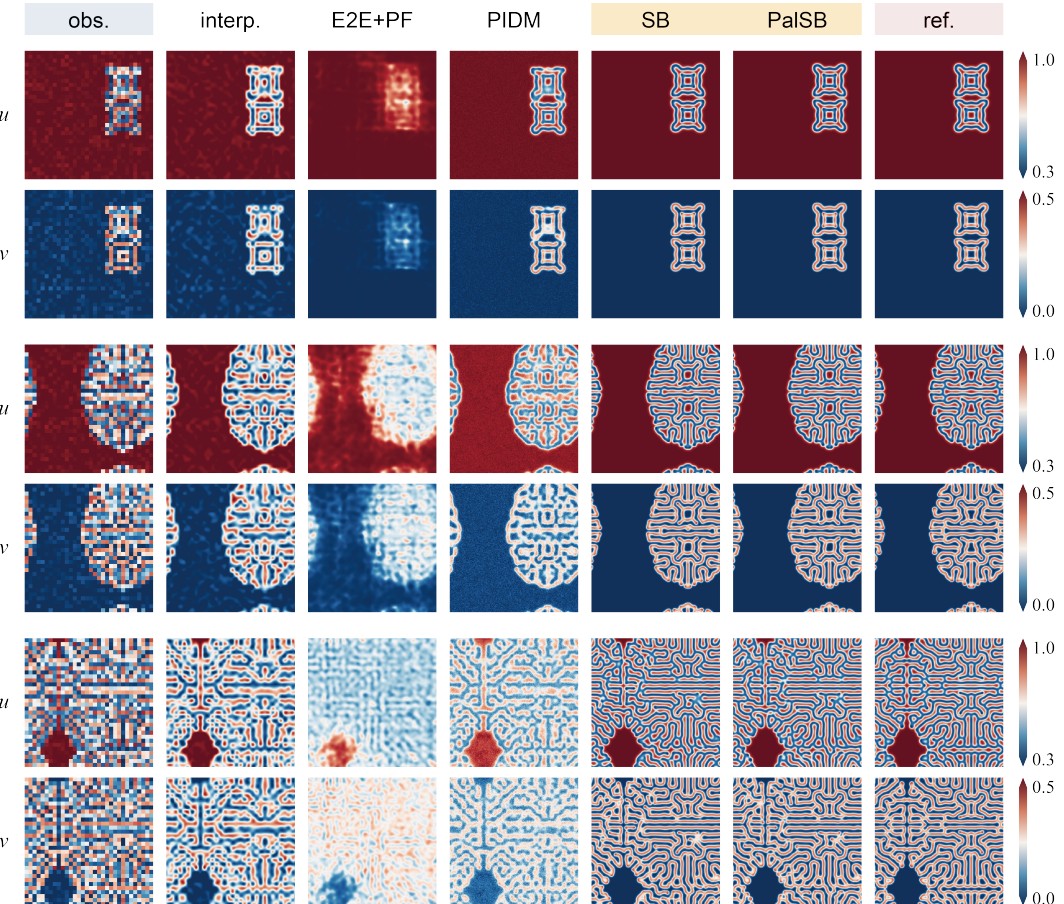

Figure 18: Additional results of GS-RD on FI task using noisy observations (8x super-resolution, 5% Gaussian noise).

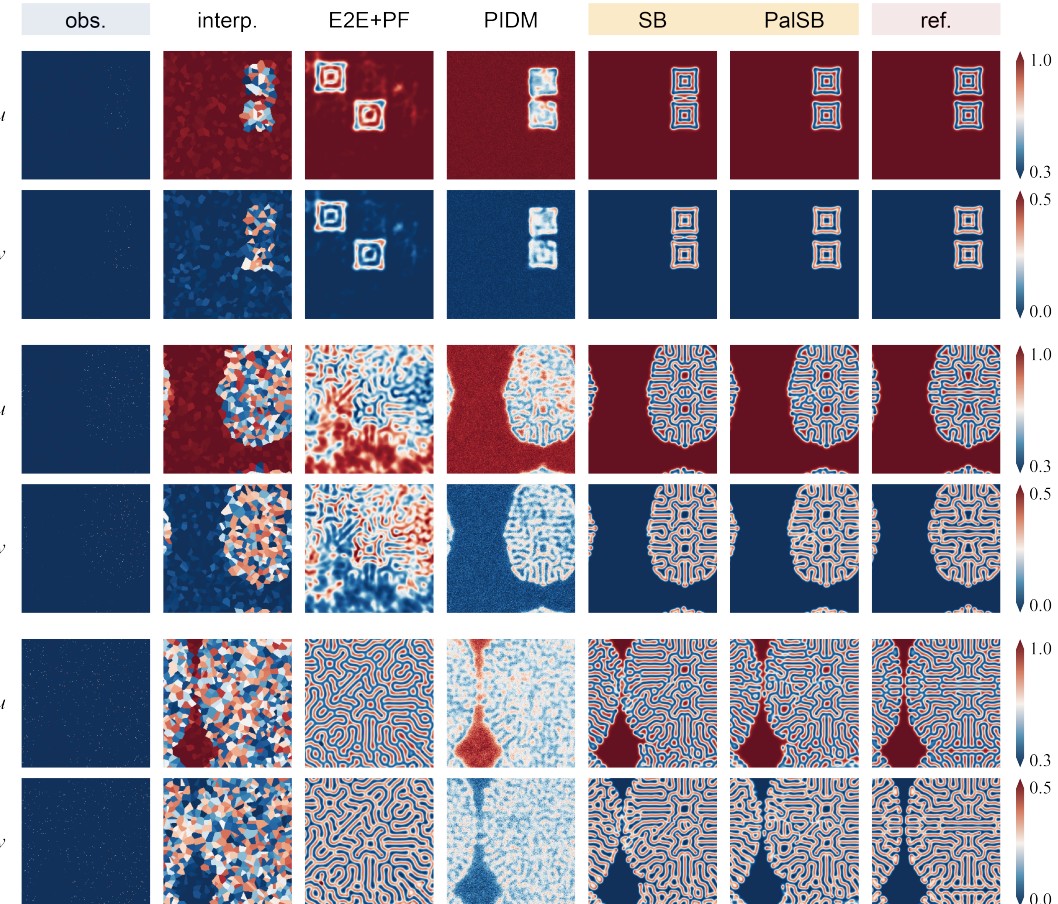

Figure 19: Additional results of GS-RD on RI task using noisy observations (99% masked, 5% Gaussian noise).

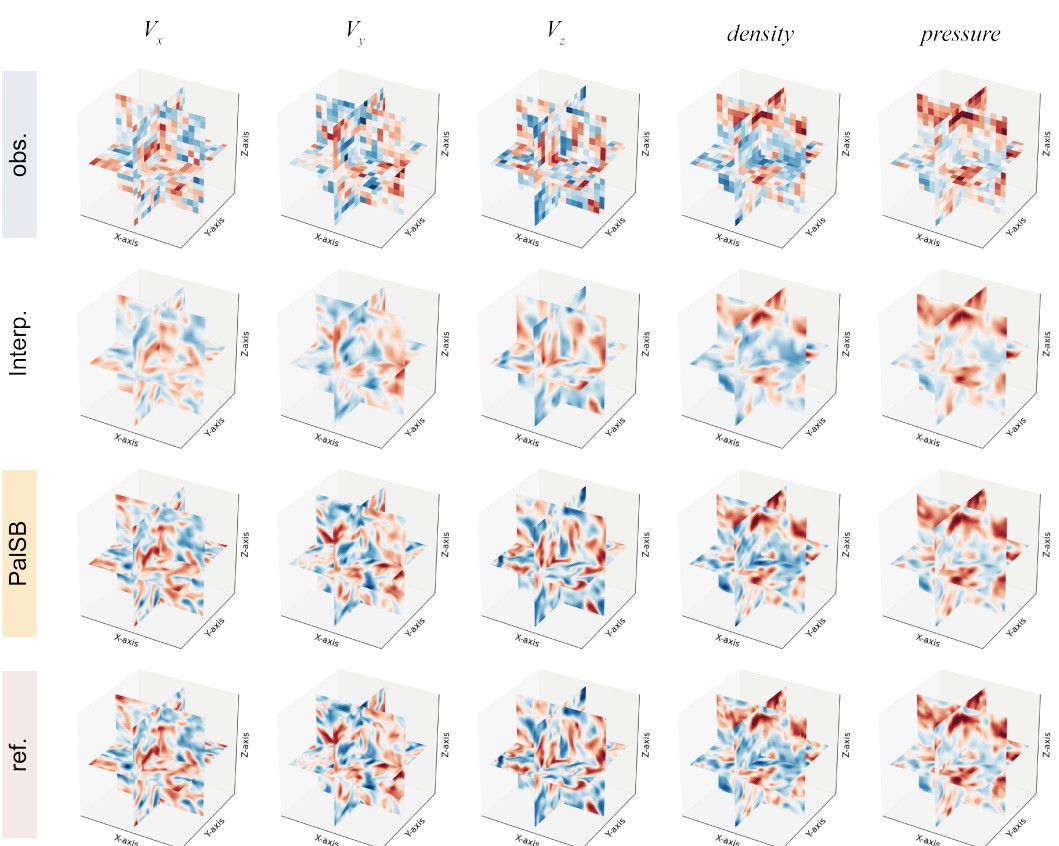

Figure 20: Extension on 3D case for FI task (4x super-resolution, noise-free).

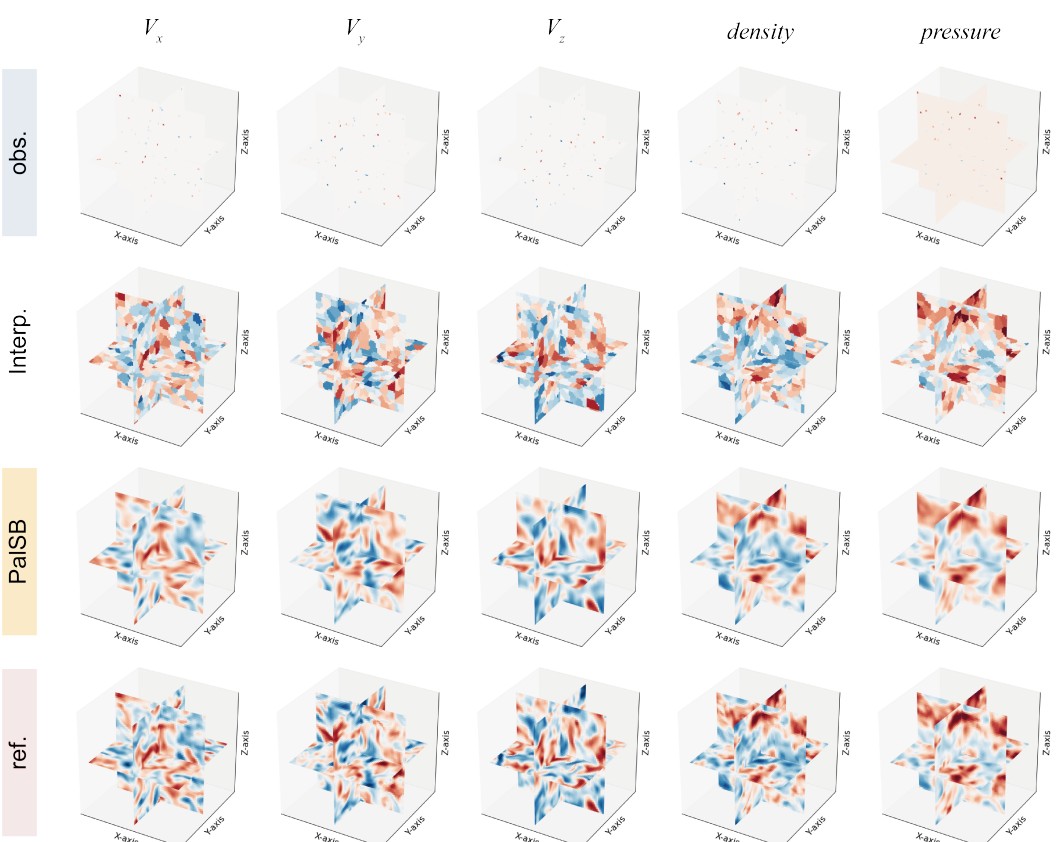

Figure 21: Extension on 3D case for RI task (99% masked, noise-free).

Table 6: Parameters of the neural network used in PalSB

| Parameter name | Cylinder flow | Kolmogorov flow | Reaction-diffusion |
|---|---|---|---|
| Model architecture | DenoisingUNet Luo et al. (2023) | DenoisingUNet | DenoisingUNet |
| $\beta_{\min}$ | 0.1 | 0.1 | 0.1 |
| $\beta_{\max}$ | 0.3 | 0.3 | 0.3 |
| Number of scales | 1000 | 1000 | 1000 |
| Positional embeds | sinusoidal | sinusoidal | sinusoidal |
| Number of features | 32 | 32 | 32 |
| Channel multiplier | (1, 2, 4, 8) | (1, 2, 4, 8, 16) | (1, 2, 4, 8, 16) |
| Number of residual blocks | 2 | 2 | 2 |
| Nonlinearity | Swish | Swish | Swish |
| Dropout | 0.1 | 0.1 | 0.1 |

Table 7: Parameters of FNO

| Parameter name | Cylinder flow | Kolmogorov flow | Reaction-diffusion |
|---|---|---|---|
| Model architecture | 2D FNO Li et al. (2020) | 2D FNO | 2D FNO |
| Modes 1 | 16 | 16 | 16 |
| Modes 2 | 16 | 16 | 16 |
| Number of features | 32 | 32 | 32 |
| Number of blocks | 4 | 4 | 4 |
| Nonlinearity | GeLU | GeLU | GeLU |

Table 8: Parameters for training

| Parameter name | Pretraining | Finetuning |
|---|---|---|
| Optimizer | AdamW | AdamW |
| Learning rate | 1e-4 | 1e-5 |
| Learning rate step | 1000 | 10 |
| Learning rate decay | 0.99 | 0.99 |
| Batch size | 64 | 32 |
| Small batch size | 64 | 4 |
| Number of iterations | 100000 | 1000 |
| Ema rate | 0.995 | - |
| Sampling steps $T$ | - | 10 |
| Sampling step size | - | 1e-3 |
| Backpropogation steps $B$ | - | 1 |

Table 9: Loss weights for finetuning

| Parameter name | Cylinder flow | Kolmogorov flow | Reaction-diffusion |
|---|---|---|---|
| $\gamma_{phys}$ | 5 | 0.5 | 1e5 |
| $\gamma_{reg}$ | 1 | 10 | 1 |

