# OpenReview forum: "Physics-aligned field reconstruction with diffusion bridge"
_ICLR.cc/2025/Conference — ICLR 2025 Spotlight_

### Official Review · Reviewer_bWQy · 2024-10-27

**Soundness:** 4
**Presentation:** 4
**Contribution:** 4
**Rating:** 8
**Confidence:** 4

**Summary:**

This paper presents the Physics-aligned Schrödinger Bridge (PalSB) method, developed for reconstructing physical fields from sparse data subject to physical constraints. Reconstruction of physical fields is a key task in fields such as fluid mechanics, meteorology, and astrophysics, where spatiotemporal information about systems must be recovered from sparse, often noisy, observations. Despite significant advances in machine learning and the use of diffusion models to solve complex data problems, many existing approaches fail to satisfy physical laws and struggle to cope with the nonlinear nature of complex physical systems.

PalSB proposes a new approach to data reconstruction that incorporates a diffusion bridge mechanism (Schrödinger Bridge) specifically adapted to satisfy physical constraints. The model is trained in two stages: the first stage focuses on local reconstruction map construction, and the second stage focuses on global physical principles. This combination allows for the creation of super-resolved data that not only more accurately but also better satisfy physical constraints than similar methods.

The pretraining stage of PalSB is based on a diffusion Schrödinger bridge, which allows for efficient transition between high-quality and corrupted data distributions. To reduce computational costs and improve the accuracy of the model, the training process is performed on small local patches of the high-resolution field. This helps the model focus on the local structure of the data, avoiding the need for large computational resources. The second stage involves Physics-aligned Finetuning, which allows the model to fit physical laws represented as differential equations. This stage minimizes deviations from physical laws using a combination of physically informed and regression loss functions.

The paper also proposes boundary-aware sampling to handle boundary conditions and early stopping sampling, which improves the generation efficiency and reduces the number of required steps. These strategies allow the model to better account for global conditions and significantly improve the performance of PalSB.

Three complex physical systems were chosen to evaluate PalSB: flow around a cylinder, two-dimensional turbulence, and a reaction-diffusion system. Based on the benchmark results, PalSB demonstrated significant improvements in accuracy and physical compliance compared to other methods, including interpolation, end-to-end models, and physically informed diffusion models (PIDM). In particular, PalSB significantly reduces physical inconsistencies and improves the compliance with spatial patterns in both Fourier and real space interpolation problems.

The study also showed that removing individual PalSB modules (e.g., physically consistent tuning or boundary-aware sampling) degrades the physical compliance and model performance. This highlights the importance of PalSB's integrated approach to achieve high accuracy and compliance with physical laws.

In conclusion, PalSB demonstrates great potential for field reconstruction in complex systems, showing high quality data recovery and improved compliance with physical constraints. In the future, PalSB is expected to be extended to 3D problems, which will cover a wider range of scientific and engineering applications that require compliance with physical laws during the generation process.

**Strengths:**

First, it provides high accuracy, surpassing traditional methods by using the diffusion bridge mechanism, which allows for efficient data recovery from sparse measurements. Second, PalSB integrates physical constraints (boundary conditions and equations) at the fine-tuning stage, which minimizes mismatches with real physical laws.

**Weaknesses:**

Unfortunately, I do not understand your motivation for using only the normalized error. I highly recommend that model errors can be estimated using the mean squared error (MSE) and L1 error metrics. In addition, the residual error, which determines how well the predicted field fits the equations describing the physical system, can be used to assess compliance with physical constraints. Also, for the future, I highly recommend combining your approach and this approach https://arxiv.org/pdf/2403.14404

**Questions:**

Can you additional explain me, why did you use only normalized metrics? Also, can you please explain why you checked on flow equations (like Kolmogorov and Reaction-Diffusion equations)? I highly recommend checking on electrostatic equations as well.

---

> ### Author Response · Authors · 2024-11-19
> **Response to reviewer bWQy**
>
> Thanks for the positive comments! We've added additional evaluations of our method using extended metrics, see following.
>
> ## Q1. Can you additional explain me, why did you use only normalized metrics?
>
> > A1:We have extended the evaluation metrics to include both MSE and L1 error metrics. The results using these metrics are included in __all the tables__ of the revised manuscript, where we observe that they indicate similar trends across our cases. For residual errors of the equations, the 'nER' metric, as presented in the manuscript, evaluates compliance with the corresponding physical constraints. For further details, please refer to __Section 4.4__ (Evaluation Metrics) of the revised manuscript.
>
> ___
>
> ## Q2. Also, can you please explain why you checked on flow equations (like Kolmogorov and Reaction-Diffusion equations)?
>
> > A2: We chose to focus on flow equations because, to our knowledge, the tasks we address (e.g., FI and RI) are more practical and relevant in scenarios involving flow equations. These equations often exhibit complex solutions with co-existing multi-scale structures, making them well-suited for applications such as climate data forecasting and downscaling [1]. While we would have liked to include a broader range of physical systems in our study, our focus was limited to fluid-related systems due to constraints in accessing domain knowledge and relevant datasets.
>
> ___
>
> References:
>
> [1] Carrassi, Alberto, et al. "Data assimilation in the geosciences: An overview of methods, issues, and perspectives." Wiley Interdisciplinary Reviews: Climate Change 9.5 (2018): e535.

---

### Official Review · Reviewer_bXrf · 2024-10-30

**Soundness:** 3
**Presentation:** 3
**Contribution:** 2
**Rating:** 6
**Confidence:** 3

**Summary:**

The paper presents a method for reconstruction of physical fields from sparse information using Schrodinger bridges. The setup uses a dual training approach where the diffusion bridge is first trained to bridge between sparse and high resolution measurements, then it is fine tuned to satisfy physical constraints. This is coupled with a special technique to care for boundary conditions. The method is tested on several examples of physical field reconstruction tasks.

**Strengths:**

- physical field reconstruction is an important task with potential high impact in applied sciences
- the presented method is well-chosen for the problem
- the method is experimentally validated on several example problems

**Weaknesses:**

- while the methodology is well chosen, I don't believe the paper presents major new methodological contributions
- some of the techniques seems somewhat adhoc, i.e. employing early stop sampling. I don't doubt that these techniques work well for the problem at hand, but they are not interesting as such from a methodological point of view
- the experimental validation is somewhat limited being restricted to 2d problems and with fairly limited datasets. A more extensive validation that clearly demonstrates the power of the method would make a stronger case for the paper, given the limited methodological contributions

**Questions:**

Can you provide strong counterpoints to the above weaknesses?

---

> ### Author Response · Authors · 2024-11-19
> **Response to reviewer bXrf**
>
> Thanks for spending time comment on this work! I hope the following responses can address your concerns.
>
> ## Q1. while the methodology is well chosen, I don't believe the paper presents major new methodological contributions.
>
> > A1:  Indeed, we borrow the idea from existing work (I$^2$SB [1] and RLHF [2, 3]) to design the two stage process while specifically tailoring it to our problem according to the characteristics of the physical field reconstruction:
> __1)__ By incorporating physics-aligned loss after the pretraining, we alleviate the optimization issue introduced by starting from a good initialization;
> __2)__ Instead of reinforcement learning in typical RLHF methods, since the physics-aligned loss is differentiable w.r.t. its input, we directly optimize the model with gradient descent truncated through the sampling path;
> __3)__ the regularization term is not simply a KL-regularization used in RLHF methods (which leverages the samples generated from the pretrained model to constrain the finetuned distribution). Instead, we constrain the finetuned distribution with the samples from pretraining data, which is more straightforward and effective in the data-driven reconstruction task.
>
> ___
>
> ## Q2. some of the techniques seems somewhat adhoc, i.e. employing early stop sampling.
>
> > A2: The motivation for employing early stop sampling stems from our observation that smaller step sizes improve sample quality due to reduced discretization error, particularly when using fewer sampling steps (e.g., 10 steps). This technique is designed to enhance efficiency by reducing the step size while maintaining a low number of sampling steps (see __Fig. 5__). Additionally, we conducted detailed ablation studies to validate the empirical performance of these techniques, demonstrating their efficacy (see __Fig. 7__ and __Section C__ of the manuscript).
>
> ___
>
> ## Q3. the experimental validation is somewhat limited being restricted to 2D problems and with fairly limited datasets.
>
> > A3: We now extended our method to a complex 3D case. The results are summarized in __Table 3__ and __Figs. 20, 21__. Please refer to __Section B.1__ of the revised manuscript for details.
>
> ___
>
> References:
>
> [1] Liu, Guan-Horng, et al. "I $^ 2$ SB: Image-to-Image Schr\" odinger Bridge." arXiv preprint arXiv:2302.05872 (2023).
>
> [2] Lee, Kimin, et al. "Aligning text-to-image models using human feedback." arXiv preprint arXiv:2302.12192 (2023).
>
> [3] Fan, Ying, et al. "Reinforcement learning for fine-tuning text-to-image diffusion models." Advances in Neural Information Processing Systems 36 (2024).

---

> > ### Comment · Reviewer_bXrf · 2024-11-19
> >
> > Thank you for the response. It is good to see the new 3D experiment. I keep my evaluation of both the strengths of the paper and the weaknesses in the methodological contribution side of the paper, and therefore maintain my score.

---

### Official Review · Reviewer_BWVW · 2024-11-03

**Soundness:** 3
**Presentation:** 3
**Contribution:** 3
**Rating:** 8
**Confidence:** 4

**Summary:**

In _Physics-Aligned Field Reconstruction with Diffusion Bridge_, the authors propose a method for reconstructing physical fields from a limited number of measurements. They combine a diffusion Schrödinger bridge, following the $I^2SB$ approach by Liu et al., with a fine-tuning step to improve consistency with physical constraints. Additionally, they adapt their sampling procedure to be "boundary-aware" by padding the input appropriately and achieve good performance with a small number of function evaluations through "early-stop sampling."

**Strengths:**

The paper addresses the important challenge of reconstructing physical fields, drawing inspiration from recent methods such as $I^2SB$ diffusion Schrödinger bridges and reinforcement learning with human feedback for diffusion models. The authors convincingly demonstrate the effectiveness of their method by evaluating it across three datasets (with two tasks each) and comparing it against several reasonable baselines. Further, they conduct an ablation study, reinforcing the value of their design choices. The early stop sampling strategy in particular could prove to be useful outside of the context of physical field reconstruction as well. The method and results are presented clearly with only minor clarifications needed.

**Weaknesses:**

There's no major weaknesses of the paper.
For minor opportunities for clarification and cleanup see Questions.

**Questions:**

Questions / Suggestions for Clarification:
1. In line 083-085: What is the relationship between $m$, $n$ and $d$?
2. I can't quite follow the derivation of eq. (6) from eq. (4), I end up with $f_\theta + x_0$ instead
3. line 268: Doesn't the backpropagation get truncated for all steps $i<N$? That's how I interpret both Fig. 1 and Alg. 2
4. Does finetuning use early stop sampling?
5. Does truncating the backpropagation in the finetuning step affect the adherence to physical constraints in a time-step dependent manner? I think this is alluded to in Appendix C1 but I think it warrants further clarification, especially any potential interaction with early stop sampling.
6. How much padding is used for sampling and does the amount of padding influence performance? Does that interact with the number of sampling steps?
7. Algorithms 2 and 3 are not totally clean:
   - 7: while $j=T$ -> while $j<T$
   - 11: $t_j$ not defined
   - 12: $t$ should instead be $t_j$

Typos/Format etc
- line 065: such diffusion bridge -> such a diffusion bridge
- missing year in reference Fan et al.
- line 329: should this say the physical field cannot be simultaneously acquired at high resolution and covering a large field of view? Surely the resolution and FOV are known.
- line 883: calculate the loss according to eq 4 -> calculate the loss according to eq 6
- line 1063: "observation mentioned above" -> vague, which one?
- Fig 10+11: noisy -> noise-free

---

> ### Author Response · Authors · 2024-11-19
> **Response to reviewer BWVW**
>
> Thanks for the positive and insightful comments from reviewer BWVW on this work! I hope the following responses can resolve the reviewer's questions.
>
> ## Q1. In line 083-085: What is the relationship between $m$, $n$ and $d$?
>
> > A1: $d$ is the spatial dimension of the physical domain (e.g., $d=2$ for 2D cases). Given a domain $\Omega \subset \mathbb{R}^d$, $n$ is the dimension of the discretized sample of data $\mathbf{x}_0$ (e.g., $n=65536$ for $256\times 256$ grid). $m$ is the dimension of the observation data $\mathbf{y}$, which is degenerated from the data sample (i.e., $\mathbf{y}=\mathcal{H}(\mathbf{x}_0)+\mathbf{\epsilon}$).
>
> ___
>
> ## Q2. I can't quite follow the derivation of eq. (6) from eq. (4), I end up with $f_\theta+\mathbf{x}_0$ instead.
>
> > A2: Thanks for the important reminder! The parameterization of the network should be
> $f_\theta=-\sigma_t \epsilon_\theta + \mathbf{x}_t $.
>  In this case, one can deviate eq. (6) from eq. (4). We have rephrased the formula in the revised manuscript to make it clearer.
>
> ___
>
> ## Q3. Line 268: Doesn't the backpropagation get truncated for all steps $i<N$? That's how I interpret both Fig. 1 and Alg. 2.
>
> > A3: There is a subscript substitution in our manuscript, resulting in the confusion. As suggested by the reviewer,  we now changed the notation for time steps into $0<t_T<t_{T-1}<...<t_1=1$ throughout the paper. Please refer to Eq. (8), Fig. 1 and Alg. 2 of the revised manuscript.
>
> ___
>
> ## Q4. Does finetuning use early stop sampling?
>
> > A4: In our experiments, the finetuning also uses early stop sampling.
>
> ___
>
> ## Q5. Does truncating the backpropagation in the finetuning step affect the adherence to physical constraints in a time-step dependent manner?
>
> > A5: In the expanded experiments, we evaluated the impact of backpropagation truncation on adherence to physical constraints. The results demonstrate that, within a fixed number of fine-tuning iterations, truncating backpropagation can enhance adherence to physical constraints without altering the trend in a time-step-dependent manner (see __Fig. 8__ in the revised manuscript). For further details, please refer to __Section C.2__ of the revised manuscript.
>
> ___
>
> ## Q6. How much padding is used for sampling and does the amount of padding influence performance? Does that interact with the number of sampling steps?
>
> > A6: According to the dimension of the observations, we use padding size of 2 for FI task while 16 for RI task to make the size of the interpolated data samples consistent. We now extended the experiments to investigate how the padding size and sampling steps affect the performance (__Fig. 9__ in the revised manuscript). The results indicate that increasing the padding size can improve the final performance without altering the trend with respect to the number of sampling steps. Please refer to __Section C.4__ in the revised manuscript for details.
>
> ___
>
> ## Q7. Issues in Algorithms 2 and 3.
>
> > A7: Thanks for the important reminder! We now rephrased Algorithms 2 and 3 of the revised manuscript to make them clearer.
>
> ___
>
> ## Typos/Format etc.
>
> > A: Thanks for pointing out the typos! We have corrected these typos in the revised manuscript.

---

> > ### Comment · Reviewer_BWVW · 2024-11-21
> >
> > Thank you for the response!
> >
> > The revisions to the manuscript have further improved the clarity and the added experiments give additional context on how to apply the method.
> >
> > One comment for Q3:
> > I agree that the new notation for the timesteps in the revised manuscript has removed the previous ambiguity. However, is there a reason to choose $0<t_T<...<t_1=1$ over the more intuitive $0<t_1<...<t_T=1$?
> >
> > I maintain my previous rating and continue to recommend this manuscript for acceptance.

---

> > > ### Author Response · Authors · 2024-11-22
> > > **Response to reviewer BWVW**
> > >
> > > Thanks for the comments!
> > >
> > > ## Q. One comment for Q3: I agree that the new notation for the timesteps in the revised manuscript has removed the previous ambiguity. However, is there a reason to choose $0 < t_T < ... < t_1 = 1$ over the more intuitive $0 < t_1 < ... < t_T = 1$?
> > >
> > > > A: We appreciate that choosing the time steps as $0 < t_1 < ... < t_T = 1$ may seem more intuitive. However, our notation ($0 < t_T < ... < t_1 = 1$) is designed to align with the sampling process, which progresses from $\mathbf{x}_1$ to $\mathbf{x}_0$. This allows the sampling process to naturally proceed as $t_1 \rightarrow t_2 \rightarrow ... \rightarrow t_T$, maintaining consistency with the direction of the evolution.

---

> > > > ### Comment · Reviewer_BWVW · 2024-12-02
> > > >
> > > > Fair enough, thank you

---

### Meta-Review · Area_Chair_Eqkv · 2024-12-16

**Metareview:**

This paper presents the Physics-aligned Schrödinger Bridge (PalSB) framework, an approach to reconstructing physical fields from sparse measurements while satisfying physical constraints (eg.governing equations and boundary conditions). PalSB employs a diffusion bridge mechanism tailored for physical systems; it uses a dual-stage training process to integrate local reconstruction accuracy with global physical compliance. The paper also introduces a boundary-aware sampling technique and an efficient early stopping strategy. The method is validated on three complex nonlinear systems (flow around a cylinder, 2D turbulence, and a reaction-diffusion system) and demonstrates improved accuracy and adherence to physical laws compared to existing methods.

The paper addresses a critical gap in current field reconstruction methods. By explicitly aligning with physical laws, PalSB significantly outperforms competing approaches (eg. interpolation, end-to-end models, and physically informed diffusion models), in both accuracy and physical compliance. Its dual-stage training, boundary-aware sampling, and early stopping strategies are validated through detailed experiments and ablation studies. The rigorous evaluation on challenging benchmarks and the potential for extending PalSB to 3D problems demonstrate its utility and applicability.

The panel of reviewers unanimously recommends acceptance, citing the paper’s strong methodological contributions, thorough experimental validation, and clear presentation. The reviewers noted that the framework has the potential to significantly advance the field of physics-compliant machine learning and have a meaningful impact on the community.

**Additional Comments On Reviewer Discussion:**

During the rebuttal period, the authors primarily focused on clarifying a few technical points raised by the reviewers and addressing minor typographical issues in the manuscript. Notably, they also extended their method to a complex 3D case, demonstrating its scalability and potential for broader applicability.

---

### Decision · Program_Chairs · 2025-01-22

Accept (Spotlight)